# Determining cellular CTCF and cohesin abundances to constrain 3D genome models

Claudia Cattoglio[1,2], Iryna Pustova[1,2†‡], Nike Walther[3†§], Jaclyn J Ho[1,2], Merle Hantsche-Grininger[3], Carla J Inouye[1,2], M Julius Hossain[3], Gina M Dailey[1], Jan Ellenberg[3], Xavier Darzacq[1], Robert Tjian[1,2], Anders S Hansen[1,2*]

[1]Department of Molecular and Cell Biology, Li Ka Shing Center for Biomedical and Health Sciences, CIRM Center of Excellence, University of California, Berkeley, Berkeley, United States; [2]Howard Hughes Medical Institute, Berkeley, United States; [3]Cell Biology and Biophysics Unit, European Molecular Biology Laboratory (EMBL), Heidelberg, Germany

*For correspondence:
anders.sejr.hansen@berkeley.edu

†These authors contributed equally to this work

Present address: ‡Department of Biomolecular Chemistry, University of Wisconsin-Madison School of Medicine and Public Health, Madison, United States; §Department of Molecular and Cell Biology, Li Ka Shing Center for Biomedical and Health Sciences, CIRM Center of Excellence, University of California, Berkeley, Berkeley, United States

**Abstract** Achieving a quantitative and predictive understanding of 3D genome architecture remains a major challenge, as it requires quantitative measurements of the key proteins involved. Here, we report the quantification of CTCF and cohesin, two causal regulators of topologically associating domains (TADs) in mammalian cells. Extending our previous imaging studies (Hansen et al., 2017), we estimate bounds on the density of putatively DNA loop-extruding cohesin complexes and CTCF binding site occupancy. Furthermore, co-immunoprecipitation studies of an endogenously tagged subunit (Rad21) suggest the presence of cohesin dimers and/or oligomers. Finally, based on our cell lines with accurately measured protein abundances, we report a method to conveniently determine the number of molecules of any Halo-tagged protein in the cell. We anticipate that our results and the established tool for measuring cellular protein abundances will advance a more quantitative understanding of 3D genome organization, and facilitate protein quantification, key to comprehend diverse biological processes.
DOI: https://doi.org/10.7554/eLife.40164.001

## Introduction

Folding of mammalian genomes into structures known as Topologically Associating Domains (TADs) is thought to help regulate gene expression while aberrant misfolding has been associated with disease (*Dekker and Mirny, 2016*; *Fudenberg and Pollard, 2019*; *Hansen et al., 2018a*; *Hnisz et al., 2017*; *Lupiáñez et al., 2015*; *Symmons et al., 2014*). CTCF and cohesin have emerged as causal regulators of TAD formation and maintenance, since acute CTCF or cohesin depletion causes global loss of most TADs (*Gassler et al., 2017*; *Nora et al., 2017*; *Rao et al., 2017*; *Wutz et al., 2017*). Concordantly, knock-out of cohesin loading proteins NIPBL (*Schwarzer et al., 2017*) and MAU2 (*Haarhuis et al., 2017*) also affect TAD organization, although to different extents. Likewise, loss of the cohesin unloader WAPL strengthens TADs (*Gassler et al., 2017*; *Haarhuis et al., 2017*; *Wutz et al., 2017*). Consistent with the key roles played by CTCF and cohesin, models of genome folding through cohesin-mediated loop extrusion, which is stopped by chromatin-bound CTCF, have been remarkably successful in reproducing the general features of genomic contact maps at the level of TADs (*Fudenberg et al., 2016*; *Fudenberg et al., 2017*; *Sanborn et al., 2015*). Nevertheless, these models have been limited by a dearth of quantitative biological data to constrain the modeling. Importantly, the number of CTCF and cohesin molecules, the molecular mechanism of loop extrusion and the stoichiometry of cohesin during this process remain unknown, further limiting

our ability to test various models. Building on our recent genomic and imaging studies of endogenously tagged CTCF and cohesin (*Hansen et al., 2017*), here we (1) estimate bounds on the density of potentially loop-extruding cohesin complexes and estimate the CTCF-binding site occupancy probability in cells; (2) provide biochemical evidence that at least a subset of cohesin complexes exist as dimers or oligomers and (3) develop a simple method for determining the absolute cellular abundance of any protein fused to the widely used and highly versatile HaloTag (*Los et al., 2008*).

## Determining the number of CTCF and cohesin proteins per cell

To estimate the absolute abundance (number of proteins per cell) of CTCF and cohesin, we applied a combination of three distinct methods: 1) 'in-gel' fluorescence, 2) Fluorescence Correlation Spectroscopy (FCS)-calibrated imaging, and 3) Flow Cytometry (FCM). First, we developed an 'in-gel' fluorescence method based on previously validated mouse and human cell lines where either CTCF (U2OS and mouse embryonic stem cells (mESC)) or the cohesin kleisin subunit Rad21 (mESC) were endogenously and homozygously Halo-tagged (*Hansen et al., 2017*). We showed that these cell lines express the tagged proteins at endogenous levels by quantitative western blotting, (*Hansen et al., 2017*). To establish a standard, we purified recombinant 3xFLAG-Halo-CTCF and Rad21-Halo-3xFLAG from insect cells and labeled the purified proteins with the bright dye $JF_{646}$ coupled to the covalent HaloTag ligand (*Grimm et al., 2015*). We then ran a known quantity of protein side-by-side with a known number of cells labeled with the same fluorescent HaloTag ligand and quantified the total protein abundance per cell using 'in-gel' fluorescence (*Figure 1A*; Materials and methods). We note that $JF_{646}$-labeling is near-quantitative in live cells (*Yoon et al., 2016*); moreover, a titration experiment indicated $\geq$90% labeling efficiency (*Figure 1—figure supplement 1A–C*), although we cannot exclude slight undercounting due to incomplete labeling. Quantification by 'in-gel' fluorescence revealed that, on average, mESCs contain ~218,000 ± 24,000 CTCF protein molecules (mean ± std) as well as ~86,900 ± 35,600 Rad21 proteins and thus presumably cohesin complexes (*Figure 1B*; Rad21 appears to be the least abundant cohesin subunit, see Materials and methods). Similarly, we determined the abundance of Halo-CTCF in U2OS cells (C32) to be ~104,900 ± 14,600 proteins per cell. The CTCF abundance in human U2OS cells corresponds thus to about half the number of CTCF molecules determined for mESCs (~218,000 proteins/cell). Independent FCS experiments in HeLa Kyoto CTCF-EGFP cells measured ~125,000 CTCF molecules per cell in G1-phase and ~181,000 in G2-phase (*Holzmann et al., 2019*). It is thus tempting to speculate that cell-type-specific control of chromatin looping may be achieved in part by regulating CTCF abundance.

Having quantified CTCF and Rad21 abundance using 'in-gel' fluorescence, we sought to test the accuracy of this method. FCS-calibrated imaging has been recently established as a robust tool for absolute protein abundance quantification (*Cai et al., 2018*; *Politi et al., 2018*; *Walther et al., 2018*). We adapted this method to Halo-tagged proteins using the commercially available HaloTag ligand TMR (*Figure 1—figure supplement 2*) and applied it to quantify cellular Halo-CTCF abundance in the U2OS C32 clone. We found a mean of 114,600 ± 10,200 CTCF proteins per U2OS interphase cell, randomly sampling asynchronously cycling single cells (mean ± std of 4 replicates with number of cells $n \geq 21$; 101 single cells in total; *Figure 1C–D*). Over 90% of cellular Halo-CTCF molecules localized to the interphase U2OS nucleus (~106,000 nuclear Halo-CTCF molecules, which corresponds to a nuclear Halo-CTCF concentration of ~144.3 nM (*Figure 1—figure supplement 2B,E*)). The result of our FCS-calibrated imaging method (~114,600 ± 10,200) agrees within technical error with our 'in-gel' fluorescence estimate of ~104,900 ± 14,600 CTCF molecules per cell, and thereby validates the latter approach for determining average cellular protein abundances. We take the mean of the two methods, 109,800 CTCF proteins per cell in U2OS cells, as our best and final cross-validated estimate.

We finally used our robust and cross-validated CTCF abundance estimate in U2OS cells as a standard to estimate protein abundances in the endogenously Halo-tagged mESC lines. We labeled cells with HaloTag TMR ligand and used FCM with TMR fluorescence as readout. After background subtraction, we could estimate the absolute abundance of TMR-labeled mESC C59 Halo-CTCF, C87 Halo-CTCF, and C45 Rad21-Halo by comparing to the standard U2OS C32 TMR-Halo-CTCF

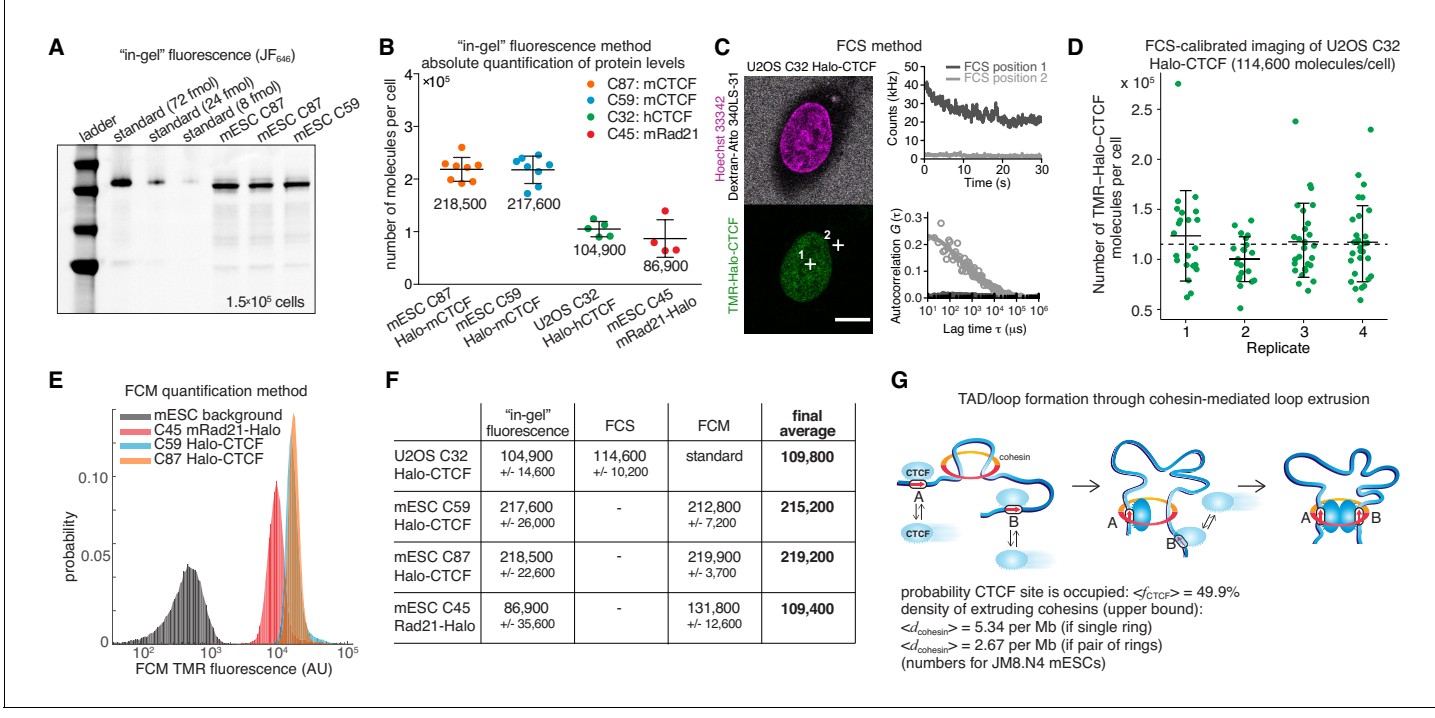

**Figure 1.** Absolute cellular CTCF and cohesin quantification. (**A**) Representative SDS-PAGE gel showing a titration of purified and labeled JF$_{646}$-3xFLAG-Halo-CTCF protein as a standard (first three lanes) side-by-side with JF$_{646}$-Halo-CTCF from lysed mESCs (3 replicates of 150,000 cells each from two different clones and different replicates). (**B**) Absolute quantification as shown in (**A**) of mESC Halo-CTCF abundance (in two independent clones, (C87 and C59), of human U2OS Halo-CTCF (clone C32) and of mESC Rad21-Halo (C45). CTCF and Rad21 were homozygously tagged in all cell lines and by western blotting the expression levels were shown to be equivalent to the untagged protein levels in wild-type cells (*Hansen et al., 2017*). Each dot represents an independent biological replicate and error bars show standard deviation. (**C**) Representative FCS measurements at points (white crosses) in the nucleus (position 1) and cytoplasm (position 2) of a U2OS Halo-CTCF C32 cell labeled with TMR HaloTag ligand. Hoechst 33342 (DNA; magenta) and Atto 340LS-31 (labeled 500 kD dextran; cell boundary marker; gray) as well as TMR (Halo-CTCF; green) channels are shown. Scale bar: 10 μm (left panel). During FCS measurements photon counts at the indicated positions (position 1, nucleus, dark gray; position 2, cytoplasm, light gray) were recorded (upper right panel) and autocorrelation curves (circles) were computed and fitted to a two-component diffusion model (lines; lower right panel; see Materials and methods for details). These FCS measurements were the basis for FCS-calibrated imaging experiments to determine the number of Halo-CTCF molecules in U2OS C32 cells as plotted in (D). See *Figure 1—figure supplement 2* and Materials and methods for details. (**D**) Four independent FCS-calibrated imaging experiments of randomly sampled interphase U2OS Halo-CTCF C32 cells labeled with TMR HaloTag ligand were performed. TMR-Halo-CTCF protein numbers were calculated for each cell (green dots; replicate 1: n = 22; replicate 2: n = 21; replicate 3: n = 29; replicate 4: n = 29). For each replicate, the mean number of TMR-Halo-CTCF molecules per cell as well as the standard deviation (error bars) are indicated. The mean calculated from the means of the four replicates is indicated as dashed line. The single-cell measurements revealed a broad distribution of Halo-CTCF abundance reflecting, amongst others, biological cell-to-cell heterogeneity of interphase cells. (**E**) Flow cytometry (FCM) quantification method. Representative replicate showing FCM-estimated TMR fluorescence of mESC lines: C45 Rad21-Halo, C59 Halo-CTCF, C87 Halo-CTCF as well as mESC background (without TMR labeling). (**F**) Table of average protein numbers per cell determined by different methods. The table provides mean ±standard deviation (std is calculated over each replicate) for each cell line and for each method. The 'final average' in bold is from averaging the different methods. (**G**) Sketch of hypothetical loop extrusion model, wherein cohesin extrudes chromatin loops until it is blocked by chromatin-bound CTCF. Below, calculation of fractional CTCF occupancy and density of extruding cohesin molecules. See Materials and methods for calculation details.

DOI: https://doi.org/10.7554/eLife.40164.002

The following figure supplements are available for figure 1:

**Figure supplement 1.** Estimating labeling efficiency and cell cycle phase distribution.
DOI: https://doi.org/10.7554/eLife.40164.003

**Figure supplement 2.** Determination of CTCF protein numbers in interphase U2OS C32 Halo-CTCF cells by FCS-calibrated-imaging.
DOI: https://doi.org/10.7554/eLife.40164.004

**Figure supplement 3.** Flow cytometry (FCM)-based quantification of mESC protein abundances.
DOI: https://doi.org/10.7554/eLife.40164.005

fluorescence (*Figure 1E*; *Figure 1—figure supplement 3*). Notably, the estimates of mESC C59 and C87 Halo-CTCF were identical within error by both the 'in-gel' fluorescence and FCM method (*Figure 1F*). We take the mean of C59 and C87 across the two methods, namely ~217,200 CTCF proteins per cell in mESCs, as the best and final estimate. This provides additional cross-validation and furthermore suggests that FCM can be used to estimate the absolute abundance of other Halo-tagged proteins if the U2OS C32 Halo-CTCF cell line is used as a standard (see below; Figure 3). For mESC C45 Rad21-Halo, the FCM estimate of 131,800 ± 12,600 proteins/cell differed more from the 'in-gel' fluorescence estimate of 86,900 ± 35,600, but was still just within error. We speculate that this discrepancy could be due to poorer Rad21-Halo protein stability during the biochemical steps of the 'in-gel' fluorescence method. We again take the mean of the two methods, ~109,400 Rad21 proteins per cell, as our final, although less certain, estimate of Rad21 abundance in mESCs.

## Quantitative constraints of 3D genome organization from CTCF and cohesin abundances

The loop extrusion model posits that cohesin extrudes chromatin loops until blocked by chromatin-bound CTCF (*Figure 1G*; *Fudenberg et al., 2017*). Based on the determined abundances of CTCF and cohesin in mESCs, we can now parameterize this model. First, we measured the interphase cell cycle distribution of JM8.N4 mESCs: 10.2% in G1-phase, 73.9% in S-phase, and 15.9% in G2-phase (*Figure 1—figure supplement 1D–G*). This approximately agrees with other mESC estimates (*Hansen et al., 2018b*; *Sladitschek and Neveu, 2015*) and shows that an 'average' mESC is approximately half-way through the cell cycle and thus contains ~3 genome copies. We have previously determined the fraction of CTCF molecules bound to specific DNA sites in mESCs by single-molecule imaging (~49%) and the total number of CTCF sites in the mESC genome by ChIP-seq (~71,000) (*Hansen et al., 2017*). Now we can use the information on the absolute abundance of CTCF proteins per mESC (217,200) to calculate that an average CTCF-binding site is occupied ~50% of the time by a CTCF molecule (always assuming three genome copies; full details in Materials and methods). In the context of the loop extrusion model, this suggests that the time-averaged occupancy of an average CTCF boundary site by CTCF is ~50% (*Figure 1G*) – that is, an extruding cohesin will be blocked ~50% of the time at an average CTCF site in the simplest version of the loop extrusion model. We cannot estimate CTCF binding site occupancy and probability of blocking cohesin extrusion in U2OS cells, since these cells have a poorly defined karyotype.

For cohesin, we previously estimated the fraction of cohesin complexes that are relatively stably associated with chromatin (~20–25 min residence time in mESC G1) and thus presumably topologically engaged to be ~40% in G1 (*Hansen et al., 2017*). If we take this as the upper bound of putatively 'loop-extruding' cohesin complexes, we can similarly calculate the upper limit on the density of extruding cohesin molecules as ~5.3 per Mb assuming cohesin exists as a monomeric ring or ~2.7 per Mb if cohesin forms dimers (*Figure 1G*; full details on calculation in Materials and methods). This corresponds to a genomic distance between extruding cohesins of ~186–372 kb in mESCs, which approximately matches computational estimates (*Fudenberg et al., 2016*; *Gassler et al., 2017*). We envision that these numbers will be useful starting points for constraining and parameterizing models of 3D genome organization and we discuss some limitations of these estimates below.

## Mammalian cohesin can form dimers and/or higher order oligomers in cells

Interpreting the cohesin data described above requires an accurate count of its molecular stoichiometry, but whether cohesin complexes function as single rings, dimers or higher order oligomeric structures (*Figure 1G*) has been highly debated in the literature. In addition to potentially engaging in loop extrusion (*Hassler et al., 2018*; *Nichols and Corces, 2018*) cohesin plays important roles in sister chromatid cohesion and DNA repair (*Guacci et al., 1997*; *Losada et al., 1998*; *Michaelis et al., 1997*; *Onn et al., 2008*). Cohesin is generally assumed to exist as a single tripartite ring composed of the subunits Smc1, Smc3 and Rad21/Scc1/Mcd1 at 1:1:1 stoichiometry (*Nasmyth, 2011*), with a fourth subunit, Scc3 (SA1 or SA2 in mammalian cells) that is bound to Rad21. However, higher order oligomeric cohesin structures have been proposed based upon the unusual genetic properties of cohesin subunits in budding yeast (*Eng et al., 2015*; *Skibbens, 2016*). Moreover, a previous study used self co-immunoprecipitation (CoIP) of cohesin subunits to suggest a

handcuff-shaped dimer model for cohesin (*Zhang et al., 2008*). Still, this study has remained highly controversial (*Nasmyth, 2011*) and self-CoIP experiments of cohesin subunits in budding yeast (*Haering et al., 2002*) and human HeLa cells (*Hauf et al., 2005*) could not detect cohesin dimers. Moreover, budding yeast condensin, an SMC complex related to cohesin, can extrude loops in vitro as a monomer (*Ganji et al., 2018*). Since the mammalian study (*Zhang et al., 2008*) relied on over-expressed epitope-tagged cohesin subunits and given our recent observations that over-expression of the Rad21 subunit does not faithfully recapitulate the properties of endogenously tagged Rad21 (*Hansen et al., 2017*), we decided to revisit this important issue using endogenous tagging without overexpression. First, we generated mESCs where one endogenous Rad21 allele was Halo-V5 tagged while the other allele was not tagged (clone C85; *Figure 2B–C*; see Materials and methods for details). We also generated an additional mESC line where one allele of Rad21 was tagged with Halo-V5 and the other with SNAP-3xFLAG (clone B4; *Figure 2B–C*). We then carefully examined the specificity of several V5 and FLAG antibodies in both western blot and CoIP assays to select those with no cross-reactivity with either the reciprocal tag or the wild-type, untagged Rad21 protein (*Figure 2—figure supplement 1A and D*). If cohesin exclusively existed as a single ring containing one Rad21 subunit, a V5 IP of Rad21-Halo-V5 should not pull down the Rad21 protein generated from the other allele. However, in the C85 clonal line, the V5 CoIP clearly precipitated wild-type Rad21 (*Figure 2D*). This cohesin:cohesin interaction appears to be protein-mediated rather than dependent on DNA association since benzonase treatment, which leads to complete DNA degradation

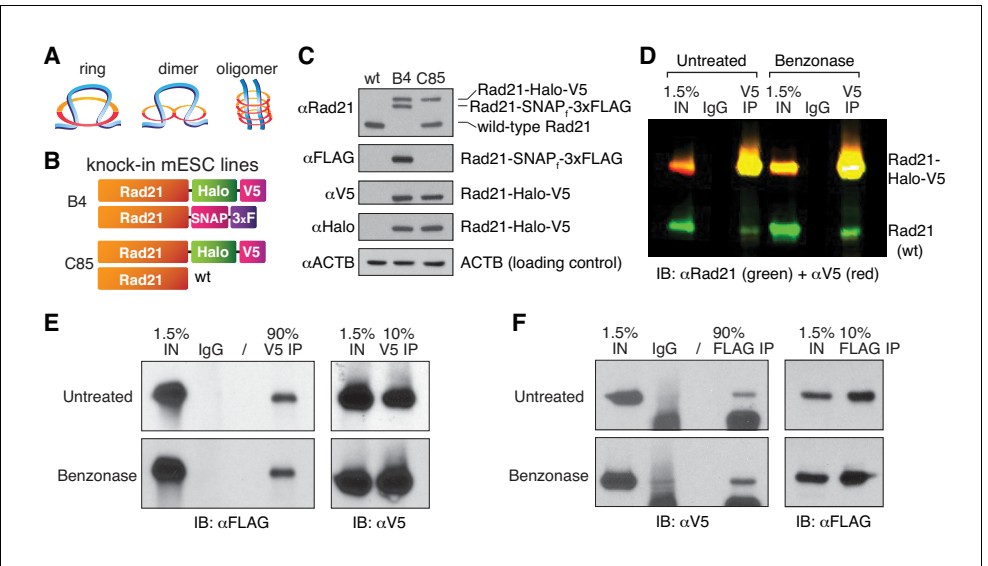

**Figure 2.** Cohesin subunit Rad21 self-interacts in a protein-dependent manner. (A) Sketches of hypothetical single-ring, dimer and oligomer models of cohesin. The core single-ring cohesin complex consists of Smc1, Smc3, Rad21 and SA1/2 subunits. (B) Schematic of Cas9-mediated, genome-edited Rad21 alleles in diploid mESCs. Clone C85 expresses Rad21-Halo-V5 from one allele and near wild-type (wt) Rad21 from the other allele (see Materials and methods for details). Clone B4 expresses Rad21-Halo-V5 from one allele and Rad21-SNAP-3xFLAG from the other. (C) Western Blot of wild-type mESCs and endogenously Rad21-tagged mESC clones shown in (B). (D) Representative CoIP experiment in mESC clone C85 indicating protein-mediated Rad21 self-interaction. V5 IP followed by two-color western blot detection with Rad21 (green) and V5 (red) antibodies shows no effect of nuclease treatment on IP and self-CoIP efficiencies. The Rad21-Halo-V5 protein reacts with both antibodies and thus appears as yellow. See also *Figure 2—figure supplement 1* for single-color blots. (E) Representative CoIP experiment in the doubly tagged B4 mESC clone. V5 IP followed by FLAG and V5 immunoblotting measures self-CoIP and IP efficiencies in the presence or absence of benzonase nuclease (90% of the IP sample loaded). (F) Reciprocal FLAG IP and quantification of benzonase DNA degradation similar to (E).
DOI: https://doi.org/10.7554/eLife.40164.006

The following figure supplement is available for figure 2:

**Figure supplement 1.** Additional blots and DNA quantification upon benzonase treatment.
DOI: https://doi.org/10.7554/eLife.40164.007

(*Figure 2—figure supplement 1C*), did not interfere with CoIP (*Figure 2D*; single-color blots in *Figure 2—figure supplement 1B*). This demonstrates that Rad21 either directly or indirectly self-associates in a protein-mediated and biochemically stable manner, consistent with cohesin forming dimers or higher order oligomers in vivo. However, this observation does not implicate that cohesin dimers or oligomers are a functional state of loop-extruding cohesin complexes.

To independently verify this result and to ensure that the CoIP'ed Rad21 was not a degradation product of the tagged protein, we repeated these CoIP studies in the clonal cell line B4, where the two endogenous Rad21 alleles express orthogonal epitope tags. Again, a V5-IP efficiently pulled down Rad21-SNAP-3xFLAG (*Figure 2E*) and, reciprocally, a FLAG-IP pulled down Rad21-Halo-V5 (*Figure 2F*). As before, the Rad21 self-interaction was entirely benzonase-resistant and thus independent of nucleic acid binding as this enzyme degrades both DNA and RNA (*Figure 2—figure supplement 1C*). Under the simplest assumption of cohesin forming dimers, we calculated that at least ~8% of cohesin is in a dimeric state during our pull-down experiment, based on our IP and CoIP efficiencies (full calculation details in Materials and methods). This percentage is likely an underestimate of the actual oligomeric vs monomeric ratio in live cells, since we expect a substantial proportion of the self-interactions not to survive cell lysis and the typically harsh IP procedures. Thus, while these results cannot exclude that some or even a majority of mammalian cohesin exists as a single-ring (*Figure 2A*), they do suggest that a measureable population may exist as dimers or oligomers. Whether this subpopulation represents handcuff-like dimers, oligomers (*Figure 2A*), cohesin clusters (*Hansen et al., 2017*) or an alternative state (e.g. single rings bridged by another factor such as CTCF) will be an important direction for future studies.

## A simple general method for determining the abundance of Halo-tagged proteins in live cells

Here, we have illustrated how absolute quantification of protein abundance can provide crucial functional insights into mechanisms regulating genome organization when integrated with genomic and/or imaging data (*Figure 1*; *Hansen et al., 2017*). The HaloTag (*Los et al., 2008*) is a popular and versatile protein-fusion platform that has found applications in a broad range of experimental systems (*England et al., 2015*). Indeed, it is currently the preferred choice for live-cell single molecule imaging. Combined with the development of Cas9-mediated genome-editing (*Ran et al., 2013*), endogenous Halo-tagging of proteins has thus become the gold standard (*Chong et al., 2018*; *Hansen et al., 2017*; *Komatsubara et al., 2019*; *Rhodes et al., 2017a*; *Rhodes et al., 2017b*; *Stevens et al., 2017*; *Teves et al., 2016*; *Teves et al., 2018*; *Youmans et al., 2018*), because it avoids the now well-established limitations and potential artifacts associated with protein overexpression (*Hansen et al., 2017*; *Shao et al., 2018*; *Teves et al., 2016*).

Now that we have determined the absolute abundance of CTCF in U2OS cells and cross-validated it using FCS-calibrated confocal imaging (*Figure 1A–D,F*), determining the absolute abundance of any other Halo-tagged protein becomes straightforward as demonstrated in *Figure 1E*: by growing a cell line homozygously encoding a Halo-tagged protein of interest side-by-side with the U2OS C32 Halo-CTCF line, absolute quantification can be achieved simply by measuring the relative fluorescence intensity using flow cytometry (*Figure 3*). To illustrate this, here we compared the background-subtracted TMR-fluorescence intensity of mESC lines carrying homozygously Halo-tagged Sox2 (*Teves et al., 2016*) and TBP (*Teves et al., 2018*) to our U2OS C32 Halo-CTCF cell line, and determined the average protein copy number per cell to be 460,517 ± 25,606 for Halo-Sox2 and 99,111 ± 29,125 for Halo-TBP (*Figure 3*; *Figure 3—figure supplement 1*). Although this method should be generally applicable, we note that it may not be robust for very lowly expressed proteins (below ~10,000 proteins per cell; *Figure 3—figure supplement 1*). Similarly, since the standard and the cell line of interest have to be measured side-by-side, the dynamic range of the flow cytometer will in principle impose an upper limit. This will be instrument-specific, but we note that our method may not be appropriate for extremely highly expressed proteins (>10–20 million proteins per cell; calculated based on the LSR Fortessa instrument used in this study). Compared to the 'in-gel' fluorescence method (*Figure 1A–B*), we believe this live-cell FCM method is both more convenient and robust, since it avoids cell lysis and other biochemical steps that may affect protein stability. The HaloTag knock-in cell lines described here will be freely available to the research community for use as a convenient standard to enable rapid absolute quantification of any Halo-tagged protein of interest.

**Figure 3.** A general and simple method for absolute quantification of cellular protein abundance. (1) Cells expressing the Halo-tagged protein of interest are grown together with one of the cell standards described here (e.g. U2OS C32 Halo-CTCF; *Figure 1B*). (2) After labeling with a fluorophore coupled to the HaloTag ligand (e.g. TMR or a JF-dye), the absolute (3) and relative (4) fluorescence intensities can be measured using flow cytometry (FCM) and thus the absolute abundance of the protein of interest can be calculated (5). Here, this is illustrated using mESC lines for Halo-Sox2 (*Teves et al., 2016*) and Halo-TBP (*Teves et al., 2018*) (raw data in *Figure 3— figure supplement 1*).
DOI: https://doi.org/10.7554/eLife.40164.008
The following figure supplement is available for figure 3:

**Figure supplement 1.** Quantification of Halo-Sox2 and Halo-TBP abundance in mESCs by flow cytometry (FCM).
DOI: https://doi.org/10.7554/eLife.40164.009

## Discussion

Despite the essential roles of cohesin in sister chromatid paring and interphase genome organization, and of condensin in mitotic chromosome compaction, the stoichiometry of these SMC complexes remains a matter of debate (*Nasmyth, 2011*; *Skibbens, 2016*). Our results suggest that a significant subpopulation of mammalian cohesin (lower bound: ~8%) may exist as either a dimer or an oligomeric complex (*Figure 2A*). This is consistent with an earlier study that relied on over-expression of tagged mammalian cohesin subunits (*Zhang et al., 2008*). Along these lines, the related bacterial SMC complex, MukBEF, also forms a dimer or and even 'dimers of dimers' (*Arciszewska et al., 2019*; *Badrinarayanan et al., 2012*; *Fennell-Fezzie et al., 2005*; *Matoba et al., 2005*; *Woo et al., 2009*). Moreover, the *B. subtilis* SMC condensin complex has been proposed to extrude DNA loops at a speed of ~50 kb/min as a dimeric handcuff complex (*Wang et al., 2017*). In budding yeast, cohesin exhibits inter-allelic complementation (*Eng et al., 2015*) consistent with a dimeric or higher order complex. However, previous self-CoIP experiments with differentially tagged budding yeast cohesin subunits failed to detect cohesin dimers or oligomers (*Haering et al., 2002*). Likewise, single-step photobleaching strongly indicates that budding yeast condensin can extrude loops as a single ring complex in a one-sided, asymmetric fashion in vitro (*Ganji et al., 2018*). Nevertheless, other studies have shown that budding yeast condensin can exist as both monomers, dimers, and oligomers and that multimeric budding yeast condensin is more active in a single-molecule magnetic tweezers-based DNA-compaction assay (*Keenholtz et al., 2017*). Furthermore, recent computer simulations suggest that only effectively two-sided extrusion (either two-sided extrusion, or one-sided extrusion with directional switching) can achieve the ~1000 fold condensin-mediated compaction observed for mammalian mitotic chromosomes (*Banigan and Mirny, 2018*). Although SMC complexes are highly conserved from prokaryotes to mammals, it remains unclear to what extent cohesin and condensin mechanistically differ and to what extent mammalian and budding yeast cohesin differ. For example, several cohesin proteins that are encoded by a single gene in budding yeast are encoded by multiple genes in mammals (e.g. Scc3 in budding yeast vs SA1 or SA2 in mammals). Since mammalian cohesin contains either SA1 or SA2, but not both (*Sumara et al., 2000*) and since SA1- and SA2-cohesin appear to mediate at least partially different functions (*Kojic et al., 2018*), one possibility would be that SA1- and SA2-cohesin might also differ in their architecture. Our CoIP results show that cohesin can exist in a dimeric and/or oligomeric state in mESCs (*Figure 2*). These oligomers may also be arising from cohesin clusters, which we previously observed with super-resolution microscopy (*Hansen et al., 2017*), or even from larger complexes that contain single ring cohesins which do not directly interact. We hope that our results here spur further investigations using orthogonal methods into the stoichiometry of mammalian cohesin and the architecture of the putatively loop-extruding cohesin complex. Moreover, although polymer-modeling of 3D genome organization is rapidly advancing (*Fudenberg et al., 2017*; *Nuebler et al., 2018*;

*Racko et al., 2018*), a paucity of quantitative data to inform us of the stoichiometry of key 3D genome organizers currently constrains our ability to test the various models that have been reported. We hope that the data presented here will prove useful in informing and advancing such efforts in the future.

The absolute CTCF and cohesin protein measurements that we report here for mESCs will be valuable to constrain current in-silico models of 3D genome organization. However, we note that these calculations have inherent limitations. First, although the different methods gave nearly identical CTCF estimates, the cohesin estimate is less certain. Second, these numbers represent averages (e.g. we averaged over different cell cycle phases, and protein abundance can vary significantly between phases of the cell cycle and even between genetically identical cells, as visible by the biological cell-to-cell heterogeneity of CTCF abundance in U2OS C32 cells determined by FCS-calibrated imaging (*Figure 1D*)). Third, although it remains unclear how ChIP-Seq peak strength relates to time-averaged occupancy, the wide distribution of CTCF ChIP-Seq read counts (*Figure 1—figure supplement 1H*) suggests that some CTCF binding sites will be occupied most of the time, while other sites are rarely bound (i.e. 50% is an average). Fourth, the density of extruding cohesin complexes is unlikely to be uniform across the genome (e.g. due to uneven loading or obstacles to cohesin extrusion by other large DNA-binding protein complexes) and our estimate is only an upper bound. Fifth, although we have previously shown that CTCF and cohesin interact as a dynamic complex (*Hansen et al., 2017*), we are currently unable to accurately estimate what fraction of chromatin-bound CTCF proteins are directly interacting with cohesin. This is an important aspect for future research, as it will constrain loop extrusion models further.

Although knowing the absolute in vivo abundance of a protein is crucial for understanding its function, methods for determining absolute protein abundances tend to be inconvenient and labor-intensive (e.g. the 'in-gel' fluorescence method in *Figure 1A–B*) and/or require extensive and sophisticated experimental and computational infrastructure (e.g. FCS-calibrated imaging (*Figure 1C–D*) or quantitative mass spectrometry (Ref: *Holzmann et al., 2019*, also submitted to eLife)). As a consequence, absolute abundance measurements are currently limited to a subset of cellular proteins (*Cai et al., 2018*; *Walther et al., 2018*). Here, we introduce and validate a simple FCM-based method using U2OS C32 Halo-CTCF as a standard for absolute protein quantification in live cells (*Figure 3*). We will freely share the cell lines described here as standards for absolute quantifications of any Halo-tagged protein of interest. Given that our FCM-based method is simple, fast and convenient, we hope that it will find widespread use for accurate quantification of absolute protein abundances.

# Materials and methods

**Key resources table**

| Reagent type (species) or resource | Designation | Source or reference | Identifiers | Additional information |
|---|---|---|---|---|
| Cell line (*Homo sapiens*) | U2OS | (*Hansen et al., 2017*) | U2OS | Wild-type U2OS cell line. RRID: CVCL_0042 |
| Cell line (*Mus musculus*) | mESC | (*Pettitt et al., 2009*) and UC Davis KOMP Repository | JM8.N4 mESC | https://www.komp.org/pdf.php?cloneID=8669 (RRID:CVCL_J962) |

*Continued on next page*

*Continued*

| Reagent type (species) or resource | Designation | Source or reference | Identifiers | Additional information |
|---|---|---|---|---|
| Cell line (*Mus musculus*) | mESC C59 Halo-CTCF | (*Hansen et al., 2017*) | mESC C59 Halo-mCTCF | mESC (JM8.N4) endogenous knock-in cell line where both endogenous copies of *Ctcf* have been N-terminally tagged with FLAG-HaloTag and where both endogenous copies of *Rad21* have been C-terminally tagged with SNAPf-V5. Clone 59 |
| Cell line (*Mus musculus*) | mESC C87 Halo-CTCF | (*Hansen et al., 2017*) | mESC C87 Halo-mCTCF | mESC (JM8.N4) endogenous knock-in cell line where both endogenous copies of *Ctcf* have been N-terminally tagged with FLAG-HaloTag. Clone 87 |
| Cell line (*Mus musculus*) | mESC C45 Rad21-Halo | (*Hansen et al., 2017*) | mESC C45 mRad21-Halo | mESC (JM8.N4) endogenous knock-in cell line where both endogenous copies of *Rad21* have been C-terminally tagged with HaloTag-V5. Clone 45. |
| Cell line (*Mus musculus*) | mESC C3 Halo-Sox2 | (*Teves et al., 2016*) | mESC C3 Halo-Sox2 | mESC (JM8.N4) endogenous knock-in cell line where both endogenous copies of *Sox2* have been N-terminally tagged with FLAG-HaloTag. Clone 3 |
| Cell line (*Homo sapiens*) | U2OS C32 Halo-CTCF | (*Hansen et al., 2017*) | U2OS C32 Halo-hCTCF | U2OS endogenous knock-in cell line where all endogenous copies of *Ctcf* have been N-terminally tagged with FLAG-HaloTag. Clone 32 |
| Cell line (*Mus musculus*) | mESC C85 Rad21-Halo-V5 het | This Paper | C85 | mESC (JM8.N4) endogenous knock-in cell line where one endogenous *Rad21* allele is Halo-V5 tagged while the other allele is 'near wild type' (see Materials and methods). Clone 85 |

Continued

| Reagent type (species) or resource | Designation | Source or reference | Identifiers | Additional information |
|---|---|---|---|---|
| Cell line (*Mus musculus*) | mESC B4 Rad21-Halo-V5/Rad21-SNAP$_f$-3xFLAG | This Paper | B4 | mESC (JM8.N4) endogenous knock-in cell line where one endogenous *Rad21* allele is Halo-V5 tagged while the other allele is SNAP$_f$-3xFLAG tagged. Clone B4 |
| Cell line (*Mus musculus*) | mESC A2 Rad21-Halo-V5/Rad21-SNAP$_f$-3xFLAG | This Paper | A2 | mESC (JM8.N4) endogenous knock-in cell line where one endogenous *Rad21* allele is Halo-V5 tagged while the other allele is SNAP$_f$-3xFLAG tagged. Clone A2 |
| Antibody | rabbit polyclonal anti-V5 | Abcam | Cat. # ab9116, RRID: AB_307024 | (1:2000) for western blot (WB) |
| Antibody | mouse monoclonal mouse anti-V5 | ThermoFisher Scietific | Cat. # R960-25, RRID: AB_2556564; | (1:5000) for WB |
| Antibody | rabbit polyclonal anti-FLAG | Sigma-Aldrich | Cat. # F7425, RRID: AB_439687 | (1:1000) for WB |
| Antibody | mouse monoclonal anti-FLAG | Sigma-Aldrich | Cat. # F3165, RRID: AB_259529 | (1:5000) for WB |
| Antibody | rabbit polyclonal anti-Rad21 | Abcam | Cat. # ab154769, RRID:AB_2783833 | (1:2000) for WB |
| Antibody | mouse anti-Rad21 | Millipore | Cat. # 05–908, RRID:AB_417383 | (1:5000) for WB |
| Antibody | mouse monoclonal anti-Halo | Promega | Cat. # G9211, RRID:AB_2688011 | (1:1000) for WB |
| Antibody | mouse monoclonal anti-βactin | Sigma-Aldrich | Cat. # A2228, RRID: AB_476697 | (1:4000) for WB |
| Peptide, recombinant protein | 3xFLAG-Halo-CTCF-His$_6$ | This Paper | 3xFLAG-Halo-CTCF-His$_6$ | See Materials and methods. |
| Peptide, recombinant protein | His$_6$-Rad21-Halo-3xFLAG | This paper | His$_6$-Rad21-Halo-3xFLAG | See Materials and methods. |
| Recombinant DNA reagent | pHTCHalo Tag | This paper | pHTCHaloTag | For FCS-calibrated imaging experiments (referred to as pHTCHaloTag), a stop codon was introduced into the pHTC HaloTag CMV-neo vector (Promega, #9PIG771) |

*Continued on next page*

*Continued*

| Reagent type (species) or resource | Designation | Source or reference | Identifiers | Additional information |
|---|---|---|---|---|
| Software, algorithm | Matlab | The Mathworks | MATLAB 2014b | https://www.mathworks.com/products/matlab.html |
| Software, algorithm | Flow cytometry analysis (Matlab) | This paper | Flow cytometry analysis | https://gitlab.com/tjian-darzacq-lab/cattoglio_et_al_absoluteabundance_2019 |
| Software, algorithm | FCSREAD (Matlab) | Mathworks File Exchange | FCSREAD (Matlab) | https://www.mathworks.com/matlabcentral/fileexchange/8430-flow-cytometry-data-reader-and-visualization |
| Commercial assay, kit | Click-iT EdU Alexa Fluor 488 Flow Cytometry Assay Kit | ThermoFisher Scientific | Click-iT EdU Alexa Fluor 488 Flow Cytometry Assay Kit | Cat. # C10425 |
| Chemical compound, drug | DAPI | Sigma-Aldrich | 4',6-Diamidine-2'-phenylindole dihydrochloride | Cat. # 10236276001 |
| Chemical compound, drug | Halo-TMR | Promega | HaloTag TMR ligand | Cat. # G8251 |
| Chemical compound, drug | Halo-JF$_{646}$ | (*Grimm et al., 2015*) | Halo-JF$_{646}$ | Please contact Luke D Lavis for distribution. |
| Chemical compound, drug | Hoechst 33342 | Sigma-Aldrich | Hoechst 33342 | Cat. # B2261 |

## Cell culture

JM8.N4 mouse embryonic stem cells (*Pettitt et al., 2009*) (Research Resource Identifier: RRID: CVCL_J962; obtained from the KOMP Repository at UC Davis) were cultured as previously described (*Hansen et al., 2017*). Briefly, mESC lines were grown on plates pre-coated with 0.1% gelatin (auto-claved and filtered; Sigma-Aldrich, G9391) under feeder-free conditions in knock-out DMEM with 15% FBS and LIF (full recipe: 500 mL knockout DMEM (ThermoFisher #10829018), 6 mL MEM NEAA (ThermoFisher #11140050), 6 mL GlutaMax (ThermoFisher #35050061), 5 mL Penicillin-streptomycin (ThermoFisher #15140122), 4.6 µL 2-mercapoethanol (Sigma-Aldrich M3148), 90 mL fetal bovine serum (HyClone FBS SH30910.03 lot #AXJ47554)). mES cells were fed by replacing half the medium with fresh medium daily and passaged every 2 days by trypsinization. Human U2OS osteosarcoma cells (Research Resource Identifier: RRID:CVCL_0042; a gift from David Spector's lab, Cold Spring Harbor Laboratory) were grown as previously described (*Hansen et al., 2017*). Briefly, U2OS cells were grown in low-glucose DMEM with 10% FBS (full recipe: 500 mL DMEM (ThermoFisher #10567014), 50 mL fetal bovine serum (HyClone FBS SH30910.03 lot #AXJ47554) and 5 mL Penicil-lin-streptomycin (ThermoFisher #15140122)) and were passaged every 2–4 days before reaching con-fluency. Both mouse ES and human U2OS cells were grown in a Sanyo copper alloy IncuSafe humidified incubator (MCO-18AIC(UV)) at 37°C/5.5% $CO_2$. Both the mESC and U2OS cell lines were pathogen-tested and found to be clean and the U2OS cell line was authenticated through STR profil-ing. Full details on pathogen-testing and authentication can be found elsewhere (*Hansen et al., 2017*).

## CRISPR/Cas9-mediated genome editing

CTCF knock-in U2OS and mESC lines were as previously described (*Hansen et al., 2017*). The Rad21 knock-in C85 and B4 mESC clones were sequentially created roughly according to published procedures (*Ran et al., 2013*), but exploiting the HaloTag and SNAPf-Tag to perform fluorescence activated cell sorting (FACS) for edited cells. The SNAPf-Tag is an optimized version of the SNAP-Tag, and we purchased a plasmid encoding this gene from NEB (NEB, Ipswich, MA, #N9183S). We transfected mESCs with Lipofectamine 3000 (ThermoFisher L3000015) according to manufacturer's protocol, co-transfecting a Cas9 and a repair plasmid (2 μg repair vector and 1 μg Cas9 vector per well in a 6-well plate; 1:2 w/w). The Cas9 plasmid was slightly modified from that distributed from the Zhang lab (*Ran et al., 2013*): 3xFLAG-SV40NLS-pSpCas9 was expressed from a CBh promoter; the sgRNA was expressed from a U6 promoter; and mVenus was expressed from a PGK promoter. For the repair vector, we modified a pUC57 plasmid to contain the tag of interest (Halo-V5 for C85 or SNAPf-3xFLAG for B4) preceded by the Sheff and Thorn linker (GDGAGLIN) (*Sheff and Thorn, 2004*), and flanked by ~500 bp of genomic homology sequence on either side. To generate the C85 Rad21-Halo-V5 heterozygous clone, we used three previously described sgRNAs (*Hansen et al., 2017*) that overlapped with the STOP codon and, thus, that would not cut the repair vector (see table below for sequences). To generate the B4 Rad21-Halo-V5/Rad21-SNAPf-3xFLAG tagged clone, we re-targeted clone C85 with sgRNAs specific to the 'near wild-type' allele (see below) while providing the SNAPf-3xFLAG repair vector.

We cloned the sgRNAs into the Cas9 plasmid and co-transfected each sgRNA-plasmid with the repair vector individually. 18–24 hr later, we then pooled cells transfected with each of the sgRNAs individually and FACS-sorted for YFP (mVenus) positive, successfully transfected cells. YFP-sorted cells were then grown for 4–12 days, labeled with 500 nM Halo-TMR (Halo-Tag knock-ins) or 500 nM SNAP-JF646 (SNAPf-Tag knock-in) and the cell population with significantly higher fluorescence than similarly labeled wild-type cells, FACS-selected and plated at very low density (~0.1 cells per mm$^2$). Clones were then picked, expanded and genotyped by PCR using a three-primer PCR (genomic primers external to the homology sequence and an internal Halo or SNAPf primer). Successfully edited clones were further verified by PCR with multiple primer combinations, Sanger sequencing and Western blotting. The chosen C85 and B4 clones show similar tagged protein levels to the endogenous untagged protein in wild-type controls (*Figure 2C*).

Genomic DNA sequencing of the C85 heterozygous clone showed the expected Halo-V5-targeted allele, and a 'near wild-type' allele, where repair following Cas9-cutting generated a 4 bp deletion (nt 2145–2148 in the NCBI Reference Sequence NM_009009.4), expected to result in a reading frame shift replacing the two most C-terminal amino acids (II) with SEELDVFELVITH. The mutation was repaired in clone B4 by providing a corrected SNAPf-3xFLAG repair vector.

All plasmids used in this study are available upon request. The table below lists the primers used for genome editing and genotyping of the Rad21 knock-in clones.

| Name/description | Sequence (5′−3′) | Experiment |
|---|---|---|
| mESC mRad21-Halo-V5 sgRNA 1: | CCTCAGATAATATGGAACCG | Genome-editing (mESC C85) |
| mESC mRad21-Halo-V5 sgRNA 2: | CCACGGTTCCATATTATCTG | Genome-editing (mESC C85) |
| mESC mRad21-Halo-V5 sgRNA 3: | ATCTAGCTCCTCAGATAATA | Genome-editing (mESC C85) |
| mESC mRad21-SNAPf-3xFLAG sgRNA 1: | AGCTCCTCAGAATGGAACCG | Genome-editing (mESC B4) |
| mESC mRad21-SNAPf-3xFLAG sgRNA 2: | TGGACCACGGTTCCATTCTG | Genome-editing (mESC B4) |
| mESC mRad21-SNAPf-3xFLAG sgRNA 3: | ACACATCTAGCTCCTCAGAA | Genome-editing (mESC B4) |
| mRad21 genome F1 | CTGGAGCACCCGTGACAGTTC | Genotyping |
| mRad21 genome R1 | CTGAGGAGTCACGCCACTGT | Genotyping |
| Internal Halo F | GTCGCGCTGGTCGAAGAATA | Genotyping (C85) |
| Internal Halo R | GGGGTCGAATGGAAAGCCA | Genotyping (C85) |
| Internal SNAPf R | CTGTTCGCACCCAGACAGTT | Genotyping (B4) |

## Antibodies

Antibodies were as follows: ChromPure mouse normal IgG from Jackson ImmunoResearch; anti-V5 for IP from Abcam (ab9116) and for Western blot (WB) from ThermoFisher (R960-25); anti-FLAG for IP (F7425) and for WB (F3165) from Sigma-Aldrich; anti-Rad21 for WB from Abcam (ab154769); anti-Halo for WB from Promega (G9211); anti-βactin for WB from Sigma-Aldrich (A2228).

## Western blot and co-immunoprecipitation (CoIP) experiments

Cells were collected from plates by scraping in ice-cold phosphate-buffered saline (PBS) with PMSF and aprotinin, pelleted, and flash-frozen in liquid nitrogen.

For western blot analysis, cell pellets where thawed on ice, resuspended to 1 mL/10 cm plate of low-salt lysis buffer (0.1 M NaCl, 25 mM HEPES pH 7.5, 1 mM MgCl2, 0.2 mM EDTA, 0.5% NP-40 and protease inhibitors), with 125 U/mL of benzonase (Novagen, EMD Millipore), passed through a 25G needle, rocked at 4°C for 1 hr and 5M NaCl was added to reach a final concentration of 0.2 M. Lysates were then rocked at 4C for 30 min and centrifuged at maximum speed at 4°C. Supernatants were quantified by Bradford. 15 μg of proteins were loaded on 8% Bis-Tris SDS-PAGE gel and transferred onto nitrocellulose membrane (Amersham Protran 0.45 um NC, GE Healthcare) for 2 hr at 100 V.

For chemiluminescent western blot detection with HRP-conjugated secondary antibodies, after the transfer the membrane was blocked in TBS-Tween with 10% milk for 1 hr at room temperature and blotted overnight at 4°C with primary antibodies in TBS-T with 5% milk. HRP-conjugated secondary antibodies were diluted 1:5000 in TBS-T with 5% milk and incubated at room temperature for an hour.

For fluorescence detection, after the transfer the membrane was blocked with the Odyssey Blocking Buffer (PBS) for 1 hr at room temperature, followed by overnight incubation at 4°C with primary antibodies in Odyssey Blocking Buffer (PBS) and PBS (1:1). IRDye secondary antibodies were used for detection at 1:5000 dilution and 1 hr incubation at room temperature. After extensive washes, the membrane was scanned with a LI-COR Odyssey CLx scanner.

For co-immunoprecipitation experiments (CoIP), cell pellets where thawed on ice, resuspended to 1 ml/10 cm plate of cell lysis buffer (5 mM PIPES pH 8.0, 85 mM KCl, 0.5% NP-40 and protease inhibitors), and incubated on ice for 10 min. Nuclei were pelleted in a tabletop centrifuge at 4°C, at 4000 rpm for 10 min, and resuspended to 0.5 ml/10 cm plate of low salt lysis buffer either with or without benzonase (600 U/ml) and rocked for 4 hr at 4°C. After the 4-hr-incubation, the salt concentration was adjusted to 0.2M NaCl final and the lysates were incubated for another 30 min at 4°C. Lysates were then cleared by centrifugation at maximum speed at 4°C and the supernatants quantified by Bradford. In a typical CoIP experiment, 1 mg of proteins was diluted in 1 ml of CoIP buffer (0.2 M NaCl, 25 mM HEPES pH 7.5, 1 mM MgCl$_2$, 0.2 mM EDTA, 0.5% NP-40 and protease inhibitors) and pre-cleared for 2 hr at 4°C with protein-G sepharose beads (GE Healthcare Life Sciences) before overnight immunoprecipitation with 4 mg of either normal serum IgGs or specific antibodies as listed above. Some pre-cleared lysate was kept at 4°C overnight as input. Protein-G-sepharose beads precleared overnight in CoIP buffer with 0.5% BSA were then added to the samples and incubated at 4°C for 2 hr. Beads were pelleted and all the CoIP supernatant was removed and saved for phenol-chloroform extraction of DNA. The beads were then washed extensively with CoIP buffer, and the proteins were eluted from the beads by boiling for 5 min in 2X SDS-loading buffer and analyzed by SDS-PAGE and western blot.

## Estimate of cohesin dimer-to-monomer ratio from CoIP experiments

Assuming that a dimeric state is responsible for the observed protein-based cohesin self-interaction, we calculated the percentage of cohesin molecules forming dimers from our CoIP experiments in the clonal cell line B4. In these cells, one allele of Rad21 is tagged with Halo-V5 and the other with SNAP-3xFLAG, and the two proteins are expressed at virtually identical levels (*Figure 2C*). We also assumed that V5:V5 and FLAG:FLAG dimers are formed with the same likelihood of V5:FLAG dimers, the latter being the only ones that our assay probes for. Since we observed no difference when treating with benzonase, we averaged all western blot results from both the V5 and the FLAG reciprocal pull-downs (*Figure 2E and F*). We used the ImageJ 'Analyze Gels' function

(*Schindelin et al., 2012*) to measure pull-down and input (IN) band intensities (*I*) and used those numbers to calculate IP and CoIP efficiencies (%) as follows:

$$\%\text{IP} = \frac{0.015 I_{\text{IP}}}{0.1 I_{\text{IN}}}$$

$$\%\text{CoIP} = \frac{0.015 I_{\text{CoIP}}}{0.9 I_{\text{IN}}}$$

with 0.015 being the percent of input loaded onto gel as a reference and 0.1 or 0.9 the amount of the pull-down material loaded onto gel to quantify the IP or CoIP efficiency, respectively.

Within the assumed scenario, we will use the V5 pull-down of *Figure 2E* to illustrate our calculations. The V5 antibody immunoprecipitates Rad21 V5 monomers ($M_{V5}$), V5:V5 dimers ($D_{V5}$), and V5:FLAG dimers ($D_{V5\text{-FLAG}}$). The %IP (i.e. the fraction of all V5 molecules that are pulled down) is thus the sum of the three terms:

%IP = $M_{V5}$+2 x $D_{V5}$ + $D_{V5\text{-FLAG}}$

where each $D_{V5}$ contains two V5 molecules, and a $D_{V5\text{-FLAG}}$ contains a single V5 molecule. Since we assumed an equal likelihood of V5 and V5-FLAG dimers, the equation becomes:

%IP = $M_{V5}$+3 x $D_{V5\text{-FLAG}}$

Since the total number of V5 and FLAG-tagged Rad21 molecules are the same:

$D_{V5\text{-FLAG}}$ = %CoIP

thus

$M_{V5}$ = %IP - 3 x %CoIP

Finally, adjusting for the efficiency of the V5 pull-down, the total percentage of Rad21 molecules in monomers can be calculated as:

% Monomeric Rad21 = $M_{V5}$ / % IP and

% Dimeric Rad21 = 1 % Monomeric Rad21

After performing the calculations described above, the resulting percentages of cohesin molecules in dimers for all the experiments were:

V5 IP, untreated: 11.23%

V5 IP, Benzonase: 7.60%

FLAG IP, untreated: 5.19%

FLAG IP, Benzonase: 6.47%

The average percentage of cohesin molecules in dimers was thus 7.62 ± 2.6% (standard deviation).

## DNA extraction and quantification

For DNA extraction, the CoIP supernatant was extracted twice with an equal volume of phenol-chloroform (UltraPure Phenol:Chloroform:Isoamyl Alcohol (25:24:1, v/v)). After centrifugation at room temperature and maximum speed for 5 min, we added the aqueous phase containing DNA of 2 volumes of 100% ethanol and precipitated 30 min at −80°C. After centrifugation at 4°C for 20 min at maximum speed, DNA was re-dissolved in 25 µl of water and quantified by nanodrop. About 100 ng of the untreated sample DNA, or an equal volume from the nuclease-treated samples, were used for relative quantification by quantitative PCR (qPCR) with SYBR Select Master Mix for CFX (Applied Biosystems, ThermoFisher) on a BIO-RAD CFX Real-time PCR system.

Primers for DNA quantification were as follows:

*Actb* promoter forward: CATGGTGTCCGTTCTGAGTGATC

*Actb* promoter reverse: ACAGCTTCTTTGCAGCTCCTTCG

## Expression and purification of recombinant 3xFLAG-Halo-CTCF and Rad21-Halo-3xFLAG

Recombinant Bacmid DNAs for the fusion mouse proteins 3xFLAG-Halo-CTCF-His$_6$ (1086 amino acids; 123.5 kDa) and His$_6$-Rad21-Halo-3xFLAG (972 amino acids; 110.2 kDa) were generated from pFastBAC constructs according to manufacturer's instructions (Invitrogen). Recombinant baculovirus for the infection of Sf9 cells was generated using the Bac-to-Bac Baculovirus Expression System (Invitrogen). Sf9 cells ($\sim 2 \times 10^6$/ml) were infected with amplified baculoviruses expressing Halo-CTCF or

Rad21-Halo. Infected Sf9 suspension cultures were collected at 48 hr post-infection, washed extensively with cold PBS, lysed in five packed cell volumes of high-salt lysis buffer (HSLB; 1.0 M NaCl, 50 mM HEPES pH 7.9, 0.05% NP-40, 10% glycerol, 10 mM 2-mercaptoethanol, and protease inhibitors), and sonicated. Lysates were cleared by ultracentrifugation, supplemented with 10 mM imidazole, and incubated at 4°C with Ni-NTA resin (Qiagen) for either 90 mins for Halo-CTCF or 16 hr for Rad21-Halo. Bound proteins were washed extensively with HSLB with 20 mM imidazole, equilibrated with 0.5 M NaCl HGN (50 mM HEPES pH 7.9, 10% glycerol, 0.01% NP-40) with 20 mM imidazole, and eluted with 0.5 M NaCl HGN supplemented with 0.25 M imidazole. Eluted fractions were analyzed by SDS-PAGE followed by PageBlue staining.

Peak fractions were pooled and incubated with anti-FLAG (M2) agarose (Sigma) and 3X molar excess fluorogenic $JF_{646}$ for 4 hr at 4°C in the dark. Bound proteins were washed extensively with HSLB, equilibrated to 0.2M NaCl HGN, and eluted with 3xFLAG peptide (Sigma) at 0.4 mg/ml. Protein concentrations were determined by PageBlue staining compared to a β-Galactosidase standard (Sigma). HaloTag Standard (Promega) was labeled according to the method described above to determine the extent of fluorescent labeling.

## Quantification of CTCF and Rad21 molecules per cell

The number of CTCF and Rad21 molecules per cell was quantified by comparing $JF_{646}$-labeled cell lysates to known amounts of purified $JF_{646}$-labeled protein standards (e.g. 3xFLAG-Halo-CTCF-$His_6$ or $His_6$-Rad21-Halo-3xFLAG) as shown in *Figure 1A*. JM8.N4 mouse embryonic stem cells (either C45 mRad21-Halo-V5; C59 FLAG-Halo-mCTCF, mRad21-$SNAP_f$-V5; or C87 FLAG-Halo-mCTCF) were grown overnight on gelatin-coated P10 plates and human U2OS osteosarcoma C32 FLAG-Halo-hCTCF cells on P10 plates. Cells were then labeled with 500 nM (final concentration) Halo-$JF_{646}$ dye (*Grimm et al., 2015*) in cell culture medium for 30 min at 37°C/5.5% $CO_2$. Importantly, it has previously been shown that Halo-$JF_{646}$ labeling is near-quantitative for cells grown in culture (*Yoon et al., 2016*). Cells were washed with PBS, dissociated with trypsin, collected by centrifugation and re-suspended in 1 mL PBS and stored on ice in the dark. Cells were diluted 1:10 and counted with a hemocytometer. Cells were then collected by centrifugation and resuspended in 1x SDS loading buffer (50 mM Tris-HCl, pH 6.8, 100 mM DTT, 2.5% beta-mercaptoethanol, 2% SDS, 10% glycerol) to a concentration of ~10,000–20,000 cells per µL. 5–8 biological replicates were collected per cell line.

Cell lysates equivalent to $5.0 \times 10^4$ to $1.5 \times 10^5$ cells were run on 10% SDS-PAGE alongside known amounts of purified $JF_{646}$-labeled 3xFLAG-Halo-CTCF-$His_6$ or $His_6$-Rad21-Halo-3xFLAG. The protein standards were processed similar to the cell lysates to account for any loss of $JF_{646}$ fluorescence due to denaturation or SDS-PAGE, allowing for quantitative comparisons. $JF_{646}$-labeled proteins were visualized on a Pharos FX-plus Molecular Imager (Bio-Rad) using a 635 nm laser line for excitation and a Cy5-bandpass emission filter. Band intensities were quantified using Image Lab (Bio-Rad). From the absolute protein standards, we calculated the fluorescence per protein molecule, such that we could normalize the cell lysate fluorescence by the fluorescence per molecule and the known number of cells per lane to determine the average number of molecules per cell.

## Fractional occupancy and mean density calculations

Next, we calculated the fractional occupancy of CTCF in JM8.N4 mouse embryonic stem cells. Previously (*Hansen et al., 2017*), using ChIP-Seq we found 68,077 MACS2-called peaks in wild-type mESCs and 74,374 peaks in C59 FLAG-Halo-mCTCF/mRad21-$SNAP_f$-V5 double knock-in mESCs. If we take the mean, this corresponds to ~ 71,200 CTCF-binding sites in vivo. This is per haploid genome. An 'average' mouse embryonic stem cell is halfway through the cell cycle and thus contains three genomes (*Figure 1—figure supplement 1D–G*). In total, an 'average' mES cell therefore contains ~ 213,600 CTCF-binding sites. Previously (*Hansen et al., 2017*), we found that 48.9% and 49.3% of Halo-mCTCF molecules were bound to cognate binding sites in the C59 and C87 cell lines (two independent clones where CTCF has been homozygously Halo-Tagged), respectively. This corresponds to a mean of 49.1%. The average number of Halo-mCTCF molecules per cell was 215,200 ± 3400 and 219,200 ± 990 in the C59 and C87 cell lines, respectively (mean across 'in-gel' fluorescence and FCM estimates ± standard deviation). This corresponds to a mean of ~ 217,200

molecules per cell. Thus, the average occupancy (i.e. fraction of time the site is occupied) per CTCF binding site is:

$$f_{\mathrm{mCTCF}} = \frac{0.491 \cdot 217200}{3 \cdot 71200} = 0.499$$

Thus, an average CTCF binding site is bound by CTCF ~ 50% of the time in mES cells. Note, that this analysis assumes that all binding sites are equally likely to be occupied. Most likely, some of the sites will exhibit substantially higher and lower fractional occupancy as suggested by *Figure 1—figure supplement 1* (i.e. some sites may be occupied essentially all of the time, whereas others only rarely).

Within the context of the loop extrusion model (*Fudenberg et al., 2016*; *Sanborn et al., 2015*), it is crucial to know the average density of extruding cohesin complexes (e.g. number of extruding cohesins per Mb). We found the average number of mRad21-Halo molecules per JM8.N4 mES cell to be ~ 109,400 ± 31,700 (mean across 'in-gel' fluorescence and FCM estimates ± standard deviation; note the significant uncertainty in this estimate). Previously (*Hansen et al., 2017*), we found 39.8% of mRad21-Halo molecules to be topologically bound to chromatin in G1 phase and 49.8% in S/G2-phase. After DNA replication begins in S-phase, cohesin adopts multiple functions other than loop extrusion (*Skibbens, 2016*). Thus, we will use 39.8% as an estimate of the upper bound of the fraction of cohesin molecules that are topologically engaged and involved in loop extrusion throughout the cell cycle. The estimated size of the inbred C57BL/6J mouse genome, the strain background from which the JM8.N4 mES cell line is derived, is 2716 Mb (*Waterston et al., 2002*). Importantly, using single-molecule tracking we found that essentially all endogenously tagged mRad21-Halo protein is incorporated into cohesin complexes (*Hansen et al., 2017*). Accordingly, we can assume that the number of Rad21 molecules per cell corresponds to the number of cohesin complexes per cell. Thus, we get an average density of 'loop extruding' cohesin complexes of (assuming again, that an 'average' cell contains three genomes):

$$d_{\mathrm{mRad21}} = \frac{0.398 \cdot 109400}{3 \cdot 2716 \text{ Mb}} = 5.34 \frac{\text{molecules}}{\text{Mb}}$$

Thus, on average each megabase of chromatin contains 5.34 loop extruding cohesin molecules. We note that it is still not clear whether cohesin functions as a single ring or as a pair of rings (*Skibbens, 2016*). Thus, if cohesin functions as a single ring, the estimated average density is 5.34 extruding cohesins per Mb and if cohesin functions as a pair, the estimated average density is 2.67 extruding cohesin complexes per Mb. We also note that it is currently unclear whether or not the density of extruding cohesins is likely to be uniform across the genome. Finally, here we have assumed that the cohesin subpopulation we observed by single-molecule live-cell imaging to be relatively stably associated with chromatin (*Hansen et al., 2017*) is entirely engaged in loop extrusion. However, this may not be the case and this estimate should therefore be interpreted as an upper bound, since the true fraction is not known.

## Flow-cytometry-based absolute abundance of Halo-tagged cell lines

To obtain the absolute abundance of the Halo-Sox2 (*Teves et al., 2016*) and Halo-TBP (*Teves et al., 2018*) cell lines, we grew them side-by-side with the U2OS C32 Halo-CTCF knock-in cell line. We labeled them with 500 nM Halo-TMR (Promega G8251) for 30 min at 37°C/5.5% $CO_2$ in a tissue-culture incubator, washed out the dye (remove medium; add PBS; remove medium; add fresh medium) and then immediately prepared the cells for Flow Cytometry. We collected cells through trypsinization and centrifugation, resuspended the cells in fresh medium, filtered the cells through a 40 μm filter and placed the live cells on ice until their fluorescence was read out by Flow Cytometry (~20 min delay). Using a LSR Fortessa (BD Biosciences) flow cytometer, live cells were gated using forward and side scattering. TMR fluorescence was excited using a 561 nm laser and emission read out using a 610/20 band pass filter. The measured mean fluorescence intensity in the C32 standard cell line was scaled to a value of 10,000 arbitrary units, and all the values measured in the other cell lines were re-scaled accordingly. Finally, the absolute abundance of protein X was obtained according to:

$$n_{\mathrm{X}} = \frac{I_{\mathrm{X}} - I_{\mathrm{mESCBackground}}}{I_{\mathrm{C32}} - I_{\mathrm{U2OSBackground}}} n_{\mathrm{C32}}$$

where $n_X$ is the absolute abundance of the protein of interest (mean number of molecules per cell), $I_X$ is the average measured fluorescence intensity of cell lines expressing protein X (in AU), $I_{Background}$ is the average measured fluorescence intensity of cell lines that were not labeled with TMR, $I_{C32}$ is the average measured fluorescence intensity of the C32 cell line standard and $n_{C32}$ is the absolute abundance of C32 (~109,800 proteins per cell).

To quantify the abundance of Sox2 and TBP in mESCs, we performed four biological replicates and the measurements for each are shown in *Figure 3—figure supplement 1*. The raw FCM data as well as the Matlab code used to analyze it is available at https://gitlab.com/tjian-darzacq-lab/cattoglio_et_al_absoluteabundance_2019 (*Hansen, 2019*; copy archived at https://github.com/elifesciences-publications/cattoglio_et_al_absoluteabundance_2019).

## Cell cycle phase analysis in mESCs

Cell cycle phase analysis was performed using the Click-iT EdU Alexa Fluor 488 Flow Cytometry Assay Kit (ThermoFisher Scientific Cat. # C10425) according to manufacturer's instructions, but with minor modifications as previously described (*Hansen et al., 2018b*). C59 mESCs (Halo-CTCF; Rad21-SNAP$_f$) were grown overnight in a six-well plate. One well was labeled with 10 μM EdU for 30 min at 37°C/5.5% $CO_2$ in a TC incubator and one well was unlabeled and used as a negative control. Cell were harvested, washed with 1% BSA in PBS, permeabilized (using 100 μl 1x Click-iT saponin-based permeabilization and wash reagent (Component D; see kit manual), mixed well and then incubated for 15 min. 0.5 ml Click-iT reaction was added to each tube and incubated for 30 min in the dark. Cells were washed with 1x Click-iT saponin-based permeabilization and wash reagent and resuspended in 1x Click-iT saponin-based permeabilization and wash reagent with DAPI (5 ng/mL) and incubated for 10 min. Cells were then spun down and re-suspended in 1% BSA in PBS and FACS performed on a LSR Fortessa Cytometer. DAPI fluorescence was excited using a 405 nm laser and collected using a 450/50 bandpass emission filter. Alexa Flour 488 fluorescence was excited using a 488 nm laser and collected using a 525/50 bandpass emission filter. Cells were gated based on forward and side scattering. Cell cycle analysis was then performed using custom-written MATLAB code as illustrated in *Figure 1—figure supplement 1D–G*. Three independent biological replicates were performed.

## JF$_{646}$-titration to estimate labeling efficiency

To estimate the efficiency of live-cell labeling of the Halo-tagged proteins, we performed a titration experiment in three biological replicates. Labeling was performed and Flow Cytometry was performed as previously described (*Hansen et al., 2017*). Briefly, mESC C59 Halo-CTCF cells were grown in a gelatin-coated six-well plate and labeled with either 0 nM, 30 nM, 100 nM, 500 nM, 1000 nM or 5000 nM Halo-JF$_{646}$ dye (*Grimm et al., 2015*) for 30 min at 37°C/5.5% $CO_2$ in a tissue-culture incubator, washed out the dye (remove medium; add PBS; remove medium; add fresh medium) and then immediately prepared the cells for Flow Cytometry. We collected cells through trypsinization and centrifugation, resuspended the cells in fresh medium, filtered the cells through a 40 μm filter and placed the live cells on ice until their fluorescence was read out by Flow Cytometry (~20 min delay). Using a LSR Fortessa (BD Biosciences) flow cytometer, live cells were gated using forward and side scattering. JF$_{646}$ fluorescence was excited using a 640 nm laser and emission read out using a 670/30 band pass emission filter. Background-corrected fluorescence was then plotted as a function of the Halo-JF$_{646}$ concentration as shown in *Figure 1—figure supplement 1A–C*. As can be seen, 500 nM Halo-JF$_{646}$ yields near-quantitative labeling in agreement with (*Yoon et al., 2016*).

## Cloning of plasmid expressing HaloTag including HaloTag linker

To generate a plasmid expressing HaloTag including HaloTag linker (referred to as pHTCHaloTag) for FCS-calibrated imaging experiments, a stop codon was introduced into the pHTC HaloTag CMV-neo vector (Promega; #9PIG771) by PCR amplification using primer A 5'-ACGTCTAGAATGCTCGAGCCAACCAC-3' and primer B 5'-ACGGCGGCCGCTTAACCGGAAATCTCC-3' (Sigma), followed by restriction digest using *Xho*I (NEB; #R0146) and *Not*I (NEB; #R0189) and subsequent ligation by T4 DNA ligase (NEB; #M0202). The plasmid was purified endotoxin-free using EndoFree Plasmid Maxi Kit (Qiagen; #12362) for transient transfection into U2OS cells. Sequence is available upon request.

## FCS-calibrated imaging and analysis of U2OS Halo-CTCF C32 interphase cells

FCS-calibrated imaging of U2OS Halo-CTCF C32 cell line was essentially performed as described (*Cai et al., 2018*; *Politi et al., 2018*; *Walther et al., 2018*).

## Cell preparation for FCS-calibrated imaging

In detail, $1.2 \times 10^4$ U2OS wild-type (WT) cells and $2 \times 10^4$ U2OS Halo-CTCF C32 cells, respectively, were seeded into individual wells (two wells for U2OS WT, one well for U2OS Halo-CTCF C32) of a Nunc eight-well LabTek #1.0 chambered coverglass (Thermo Fisher Scientific; #155411) 2 days before imaging and incubated at 37°C and 5% $CO_2$ in a cell culture incubator. On the following day, in one-well U2OS WT cells were transiently transfected with 150 ng pHTCHaloTag plasmid using FuGENE6 Transfection Reagent (Promega; #E2693) according to the manufacturer's instructions. On the day of imaging, cells were labeled with 500 nM HaloTag TMR ligand (Promega; #G8252) in cell culture medium for 30 min at 37°C and 5% $CO_2$ in a cell culture incubator. Cells were washed with PBS and incubated in cell culture medium for 10 min at 37°C and 5% $CO_2$. Cells were again washed with PBS and 250 µl imaging medium ($CO_2$-independed imaging medium without phenol red; custom order based on #18045070 from Thermo Fisher Scientific; supplemented with 10% v/v FBS (Thermo Fisher Scientific; #10270106; qualified, European Union approved, and South American origin), 1 mM sodium pyruvate (Thermo Fisher Scientific; #11360070) and 2 mM L-glutamine (Thermo Fisher Scientific; #25030081)) was added per well containing 1 µg/ml Bisbenzimide Hoechst 33342 (Sigma-Aldrich; #B2261) and in addition for U2OS Halo-CTCF C32 cells 2 µM 500-kD Dextran (Thermo Fisher Scientific; #D7144) labeled with Atto 430LS-31 (Molecular Probes; #AD-430LS-31; Dextran-Atto 430LS-31 was produced in house; *Politi et al., 2018*).

## FCS-calibrated imaging

FCS measurements and fluorescence images were recorded on a Zeiss LSM780, Confocor3, laser scanning microscope equipped with a fluorescence correlation setup and a temperature control chamber. Imaging was performed at 37°C and using a C-Apochromat UV-visible-IR 40X/1.2-NA water objective lens (Zeiss). Data acquisition was performed using ZEN 2012 Black software (Zeiss) as well as in-house developed software applications (*Politi et al., 2018*). An in-house-designed objective cap and a water pump enabled automatic water immersion during data acquisition.

To determine the effective confocal volume, FCS measurements of a 50 nM fluorescent dye solution containing an equimolar mix of Alexa Fluor 488 (Thermo Fisher Scientific; #A20000) and Alexa Fluor 568 (Thermo Fisher Scientific; #A20003) were carried out using the 488 nm laser (laser at 0.2% excitation (exc.) power) and the 561 nm laser (laser at 0.06% exc. power) and avalanche photodiode (APD) detectors with band pass filters (BPs) set to 505–540 nm and 600–650 nm, respectively. Photon counts were recorded for 30 s and six repetitions were performed. For all three cell samples, namely WT U2OS cells, WT cells transiently transfected with pHTCHaloTag plasmid to express free HaloTag and U2OS Halo-CTCF C32 cells, all labeled with HaloTag TMR ligand, single plane images (*xy* pixel size 200 nm; image size 512 × 512 pixels; pixel dwell time 0.79 µs; 4x line averaging) were recorded with gallium arsenide phosphide (GaAsP) detectors in the TMR (561 nm laser, laser at 1% exc. power, detection window at 571–695 nm) and Hoechst 33342 (405 nm laser, laser at 0.2% exc. power, detection window at 410–481 nm) channels in separate tracks using main beam splitters (MBS) at 458/561 nm and 405 nm, respectively as well as in the transmission channel. For U2OS Halo-CTCF C32 cells, the Atto 430LS-31 channel (458 nm laser, laser at 6.0% exc. power, detection window at 491–553 nm) was additionally recorded in a separate track. In addition to a single plane image, two FCS measurement points were set per cell, one inside the nucleus and one inside the cytoplasm, and photon counts were recorded using the 561 nm laser (laser at 0.06% exc. power) and the APD detector (BP 600–605 nm) for 30 s per measurement point. To determine background fluorescence and background photon counts, FCS measurements were performed in WT U2OS interphase cells labeled with HaloTag TMR ligand. To estimate an experiment-specific calibration factor used to transform HaloTag-TMR fluorescence into HaloTag-TMR concentration, FCS measurements were performed in WT U2OS interphase cells transiently expressing different levels of free HaloTag labeled with HaloTag TMR ligand as well as in cells expressing Halo-CTCF labeled with HaloTag TMR ligand.

High-resolution confocal images covering the whole volume of individual interphase U2OS cells homozygously expressing Halo-CTCF labeled with HaloTag TMR ligand were acquired as described above for the single plane images for FCS calibration, whereby z-stacks consisting of 21 planes with a z interval of 600 nm were recorded.

## Analysis of FCS-calibrated imaging data

FCS data processing and generation of calibrated images was performed as described (*Cai et al., 2018*; *Politi et al., 2018*; *Wachsmuth et al., 2015*; *Walther et al., 2018*). To reconstruct chromosomal and cell surfaces from the Hoechst 33342 (DNA) and Dextran-Atto 430LS-31 (cell boundary) channels, respectively, a previously developed 3D segmentation pipeline (*Cai et al., 2018*; *Walther et al., 2018*) was optimized for U2OS interphase cells. In detail, in order to reduce the processing time, the original z-stack was cropped so that only the central 72 µm x 72 µm xy region of the stack remained. Cropped stacks were interpolated along the z direction to generate isotropic stacks from anisotropic source data and a 3D Gaussian filter was applied. The nuclear mass was detected from the Hoechst 33342 channel by applying adaptive thresholding (Otsu) on each xy plane of a z-stack as well as on all xy planes from the stack together (*Hériché et al., 2014*). The volume and the number of the detected binary masses were compared with a range of values determined empirically to accept the detected threshold. Otherwise, re-thresholding was performed iteratively after suppressing the higher intensity values in the histogram. Morphological features of individual connected components were analyzed to merge or split the components and binary masses with very small volumes were excluded from further processing. The remaining masses were utilized as markers to detect individual cell regions from the Dextran-Atto 430LS-31 channel using a marker-based watershed algorithm. The volumes of individual nuclear masses and their distances from the center of the image were used to detect the nuclear as well as the cell mass of interest.

The segmentation of cell and nuclear masses allowed the determination of several parameters, such as volume and total fluorescence intensity, in the whole cell as well as in the nuclear and cytoplasmic compartments. These parameters were used to calculate the average concentration of Halo-CTCF proteins and their total number in each of these compartments according to *Politi et al. (2018)*, by using the following equations:

$$C_a = \left(\frac{I_t}{V_p} - I_b\right) * k_{nM} \tag{1}$$

$$N_t = \left(I_t - V_p * I_b\right) * k_{nM} * V_\mu * N_A \tag{2}$$

where $C_a$ is the average concentration of the corresponding compartment, $I_t$ is the total intensity, $V_p$ is the total volume in number of pixels, $I_b$ is the background intensity, $k_{nM}$ is the FCS calibration factor in nM (nmol/L), $N_t$ is the total number of proteins and $V_\mu$ is the total volume in µm. $N_A$ is derived from the Avogadro constant and set to 0.602214086 so that the units equal out.

In total, four independent FCS-calibrated imaging experiments were performed, whereby the number of cells was $n \geq 21$. Mean and standard deviation of the number of TMR-labeled Halo-CTCF molecules per U2OS interphase cell were calculated per experiment as well as from all four replicates using Excel (2007; Microsoft).

## Code and data deposition

The source code for 3D segmentation of cellular and nuclear compartments of interphase U2OS cells to determine their volumes as well as for calculating protein concentrations and protein numbers within these compartments based on a FCS calibration curve (*Politi et al., 2018*) is available at https://git.embl.de/grp-ellenberg/genome_organization_cattoglio_2019. Confocal z-stacks of TMR-labeled U2OS Halo-CTCF cells, FCS calibration curves, and a summary results table are deposited at BioStudies database (*McEntyre et al., 2015*) under the accession number S-BSST229.

## Acknowledgements

We thank Luke Lavis for generously providing JF dyes, Sheila Teves for sharing cell lines, and Doug Koshland, Davide Pietrobon, Hugo Brandao and members of the Tjian-Darzacq lab for insightful

discussion and comments on the manuscript. ASH acknowledges support from a Siebel Stem Cell Institute post-doctoral fellowship and NIH NIGMS K99 Pathway to Independence Award K99GM130896. Work in the Tjian-Darzacq lab was supported by NIH Common Fund 4D Nucleome Program U01-EB021236 and U54-DK107980 (XD), the California Institute of Regenerative Medicine grant LA1-08013 (XD), and by the Howard Hughes Medical Institute (003061, RT). Work in the Ellenberg lab was supported by the European Molecular Biology Laboratory (NW, MHG, MJH, JE) and grants by the EU-H2020-iNEXT (Grant Agreement 653706), National Institutes of Health Common Fund 4D Nucleome Program (U01 EB021223/U01 DA047728) and Allen Distinguished Investigator Program, through The Paul G Allen Frontiers Group, all to JE. NW was furthermore supported by the EMBL International PhD Programme (EIPP).

## Additional information

### Competing interests

Robert Tjian: member of eLife's Board of Directors. The other authors declare that no competing interests exist.

### Funding

| Funder | Grant reference number | Author |
| --- | --- | --- |
| Siebel Stem Cell Institute | | Anders S Hansen |
| Howard Hughes Medical Institute | 003061 | Robert Tjian |
| National Institutes of Health | 4D Nucleome Common Fund UO1-EB021236 | Xavier Darzacq |
| National Institutes of Health | 4D Nucleome Common Fund U54-DK107980 | Xavier Darzacq |
| California Institute of Regenerative Medicine | LA1-08013 | Xavier Darzacq |
| National Institutes of Health | K99GM130896 | Anders S Hansen |
| European Molecular Biology Laboratory | | Nike Walther<br>Merle Hantsche-Grininger<br>M Julius Hossain<br>Jan Ellenberg |
| National Institutes of Health | 4D Nucleome Common Fund U01 EB021223 | Jan Ellenberg |
| National Institutes of Health | 4D Nucleome Common Fund U01 DA047728 | Jan Ellenberg |
| The Paul G Allen Frontiers Group | Allen Distinguished Investigator Program | Jan Ellenberg |
| Horizon 2020 Framework Programme | iNEXT 653706 | Jan Ellenberg |
| European Molecular Biology Laboratory | EMBL International PhD Programme (EIPP) | Nike Walther |

The funders had no role in study design, data collection and interpretation, or the decision to submit the work for publication.

### Author contributions

Claudia Cattoglio, Conceived and supervised the project, Designed the experiments, Designed and performed genome-editing and cohesin co-IP experiments, Drafted and edited the manuscript; Iryna Pustova, Designed and performed genome-editing and cohesin co-IP experiments, Edited the manuscript; Nike Walther, Designed, performed and analyzed FCS-calibrated imaging experiments; Jaclyn J Ho, Carla J Inouye, Performed absolute protein abundance quantification, Edited the manuscript; Merle Hantsche-Grininger, Designed, performed and analyzed FCS-calibrated imaging

experiments, Cloned HaloTag-expression plasmid; M Julius Hossain, Optimized 3D cell segmentation pipeline tailored to U2OS cells; Gina M Dailey, Generated BacMids, Edited the manuscript; Jan Ellenberg, Supervised FCS-calibrated imaging experiments; Xavier Darzacq, Robert Tjian, Conceived and supervised the project, Edited the manuscript; Anders S Hansen, Conceived and supervised the project, designed the experiments, Performed absolute protein abundance quantification, Performed flow cytometry and Sox2 and TBP abundance quantification, Drafted and edited the manuscript

## Author ORCIDs

Claudia Cattoglio https://orcid.org/0000-0001-6100-0491
Nike Walther https://orcid.org/0000-0002-7591-5251
M Julius Hossain https://orcid.org/0000-0003-3303-5755
Jan Ellenberg https://orcid.org/0000-0001-5909-701X
Xavier Darzacq https://orcid.org/0000-0003-2537-8395
Robert Tjian https://orcid.org/0000-0003-0539-8217
Anders S Hansen https://orcid.org/0000-0001-7540-7858

## Decision letter and Author response
Decision letter https://doi.org/10.7554/eLife.40164.016
Author response https://doi.org/10.7554/eLife.40164.017

# Additional files

## Supplementary files
• Transparent reporting form
DOI: https://doi.org/10.7554/eLife.40164.011

## Data availability

The raw FCM data as well as the Matlab code used to analyze are available at https://gitlab.com/tjian-darzacq-lab/cattoglio_et_al_absoluteabundance_2019 (copy archived at https://github.com/elifesciences-publications/cattoglio_et_al_absoluteabundance_2019 ). The source code for 3D segmentation of cellular and nuclear compartments of interphase U2OS cells to determine their volumes as well as for calculating protein concentrations and protein numbers within these compartments based on a FCS calibration curve (Politi et al., 2018) is available at https://git.embl.de/grp-ellenberg/genome_organization_cattoglio_2019. Confocal z-stacks of TMR-labelled U2OS Halo-CTCF cells and FCS calibration curves are deposited at BioStudies database (McEntyre et al., 2015) under the accession number S-BSST229.

The following previously published dataset was used:

| Author(s) | Year | Dataset title | Dataset URL | Database and Identifier |
|---|---|---|---|---|
| Sejr Hansen A, Cattoglio C, Pustova I, Tjian R, Darzacq X | 2016 | Nuclear organization and dynamics of CTCF and cohesin | http://www.ncbi.nlm.nih.gov/geo/query/acc.cgi?acc=GSE90994 | NCBI Gene Expression Omnibus, GSE90994 |

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
