## [Decision Letter]

Thank you for submitting your article "Architectural Features of 3D Genome Organization Revealed by Counting CTCF and Cohesin Molecules" for consideration by *eLife*. Your article has been reviewed by three peer reviewers, including David J Sherratt as the Reviewing Editor and Reviewer #1, and the evaluation has been overseen by Kevin Struhl as the Senior Editor.

The reviewers have discussed the reviews with one another and the Reviewing Editor has drafted this decision to help you prepare a revised submission.

The manuscript provides quantitative abundance estimates of the cohesin subunit Rad21 and of the architectural protein CTCF. When validated these are invaluable to the chromatin organization field and for the modelling of genome folding and loop extrusion processes. The manuscript also proposes novel methods to quantify absolute protein abundancy in living cells. Nevertheless, better validation of the methods and discussion of their limitations is required. In places the manuscript contains 'over-selling' and 'over-interpretation', combined in places with a lack of clarity and failure to discuss the field incisively.

Our agreed consensus is to ask you for a revised submission based on the comments below. All three of us are of the opinion that there is potentially important and useful material in the manuscript, but that significant revision is required if it is to be published in *eLife*.

Clarity/over-interpretation

1) Regarding the Title of the manuscript, we didn't see which 'Architectural features' have been revealed by the work. 'Quantitative features…' would be better. Furthermore, the work does not address directly '3D organisation'. A simple accurate title would do the work justice.

2) In the Introduction, the authors don't quite correctly cite the recent work on cohesin depletion. Schwarzer et al., did not deplete cohesin itself, but like Haarhuis et al., rather depleted a subunit of the cohesin loader complex. Both these papers should be cited appropriately.

3) One reviewer felt that 'CTCF boundary permeability' is an inappropriate term for the Abstract without explanation of what this esoteric term means. A second reviewer was of the opinion that the work did not directly address CTCF boundary permeability as stated in the Abstract. The reviewer states: 'to determine boundary permeability, one would need to do either in vitro experiments that assess cohesin passage, or otherwise perform detailed genomic analysis of TAD boundary strength; experiments beyond the scope of the current manuscript.

4) Similarly, the reviewers were concerned that the absolute numbers of Rad21 taken together with the fraction that was stably bound (previous work) was distilled into a statement concerning 'the number of cohesins that are extruding loops'. We do not understand the basis of this statement, since 'loop extrusion' is not assayed or directly addressed in the manuscript. Stably DNA-bound cohesin could be doing anything, and in other cases maybe even be doing nothing (e.g. if it stops translocating at appropriately paired CTCF-bound CTCF sites). Given the data, it is premature to make assumptions on the amount of extruding cohesin complexes. The authors work beautifully determines the upper limit for this number, as we now know that are no more than a certain number of complexes per cell. The authors should phrase the text accordingly and should not say that they have 'determined the density of extruding cohesins', as they do in the Abstract and in the main text.

5) One thing that is missing is determination of (or discussion of) the fraction of cohesin molecules that are associated with CTCF at CTCF sites (wasn't this information potentially accessible using the methods here; for example, using co-immunoprecipitation?).

6) Overall, the manuscript needs finessing with care to avoid over-interpretation and repetition (e.g. the statement 'In the context of the loop extrusion model, is repeated at least twice). In places it is not clear whether bound CTCF or bound cohesin is being talked about; for example, in subsection “Absolute CTCF/cohesin quantification and implications for 3D genome organization”, occupancy by what? (I presume CTCF?) – this needs to be stated explicitly.

Cohesin oligomers?

This is a contentious and much debated issue, except for the *E. coli* SMC complex MukBEF, in which the dimeric kleisin provides the basis for the inferred dimer of dimer complexes in vivo. The authors show by co-immunoprecipitation experiments of endogenously tagged Rad21 that a significant fraction of Tagged Rad21 can form dimers or multimers (assessed by co-immunoprecipitation with unlabelled or differentially tagged Rad21) or be in complexes with other proteins, where separate and potentially non-interacting cohesin monomers are co-immunoprecipitated?. The statement: 'this demonstrates that Rad21 either directly or indirectly self-associates in a protein-mediated and biochemically stable manner consistent with dimers or higher order oligomers' is just about OK, although the authors veer towards a model that this likely means functional cohesin dimers, when as best as can be ascertained this could have resulted from single rings interacting independently with other proteins (CTCF?). The issue needs better and more incisive discussion that takes into account other published work. Haering et al., (2002, Figure 6) previously found that budding yeast cohesin rarely if ever forms dimers. They also used endogenously tagged alleles. How do the authors explain their own quite different results? A potential scenario is that the IP used in the current manuscript unspecifically pulls along Rad21. A good control would therefore be to side by side include the untagged parental cells. This should yield no Rad21 pull-downs. Furthermore, Ganji et al., (2018) recently found that the related condensin complex extrudes loops as monomers. Considering the structural similarity between cohesin and condensin, it seems intuitively more likely that cohesin also acts as a monomeric complex. The authors should explain how they envisage that cohesin then acts as a dimer, while condensin acts as a monomer. One possibility could for example be that cohesin dimers/multimers do sometimes exist, but that they play no role in loop formation-perhaps they are found at stalled cohesin-CTCF complexes? If the authors wish to make a strong case for the dimer hypothesis, it would be necessary to additionally show cohesin dimers by using a different method, particularly since oligomerization has only been probed on V5-tagged protein. Therefore, repeating the experiments with a different couple of tags, or probing oligomerization by a different method could strengthen this point of the manuscript. Could the authors use fluorescence fluctuation measurements (e.g. Numbers and Brightness analysis or Photon Counting Histogram analysis) on Halo-Tagged cohesin? This could also provide quantitative estimates on the fraction of cohesin molecules engaged in oligomers.

Validation and assumptions

1) In subsection 'A simple general method for counting Halo-tagged proteins in living cells', the authors do not provide sufficient data supporting the reliability of the proposed method. The text does not discuss that abundance of proteins can change throughout the cell cycle. Therefore, the method has the same limitation as other ensemble methods, unless cells are sorted into different cell cycle stages, or single-cell microscopy is used-this could be a useful discussion point?

2) At a minimum, the authors should apply the method described in Figure 3 to compare the copy number of two proteins that they already measured with another method. For example, the authors could use the c45 (mRAD21-Halo) cell line as a standard, and apply the method of Figure 3 to the c87 (Halo-CTCF) cell line, in order to verify that the "in-gel fluorescence" and the "live-cell" method can provide comparable estimates for the number of CTCF molecules/cell. Also, it is not clear from the text whether the authors tried to quantify the *Sox2* protein number by both flow-cytometry and by microscopy. If yes, the authors should mention how the estimated protein number by these two methods differ. If not, they should only describe the experiments they actually did, and maybe comment in the Discussion section about the possible extension of the proposed method using other approaches (e.g. cell cytometry vs. live microscopy) – also considering that the dynamic range of these two methods is expected to be different.

3) Additional control experiments should be performed in order to test the reliability of the novel methods introduced and applied in this manuscript. For example, in the text that relates to Figure 1, the authors use the fluorescent signal from a bright fluorescent ligand (JF646) to quantify the protein copy number of endogenous proteins tagged with HaloTag via genome editing. The authors claim that previous work has shown that JF646 labelling is "quantitative" in living cells, but it seems that in the cited paper (Yoon et al., 2016) the fluorescent ligand is only used in a quantitative way to estimate relative changes in expression, with no real proof of its reliability for estimating absolute numbers of proteins. Importantly, the authors should cross-validate the proposed method using an established approach in the same cell line, for example by fluorescence correlation spectroscopy or mass spectrometry. At least the authors could show what are the minimum and maximum protein copy numbers measured in the cellular population and how they relate with the dynamic range in endogenous protein expression measured for example by immunofluorescence or fluorescence cytometry on the non-labelled protein population.

4) Did the authors check if the HaloTag sequence interfere with transcription, RNA stability, protein synthesis, protein degradation? Any of these processes can influence the protein copy number. At minimum the authors should show by quantitative Western Blotting that the amount of HaloTag-CTCF in the gene-edited cell line is identical to the amount of non-tagged CTCF in the parental cell-line.

5) How can the authors be sure that all the protein is extracted and detected following cell lysis? How can the authors be sure that all the protein is labelled by the JF646 fluorescent ligand? Interestingly the authors used 500nM ligand to label cellular CTCF, while a previous paper from another group using HaloTag labelling (Mazza et al., 2012 – Supplementary Figure 1) estimated that at this concentration only about 80% of a relatively low abundancy protein such as p53 is labelled by a similar concentration of TMR. The authors could titrate the concentration of fluorescent ligand added to the cells to identify a saturating concentration. The same concerns could potentially apply whenever the methods would be used on a different protein or in a different cell line. For example, the membrane permeability to the ligand might be cell-dependent and the degradation rate of different proteins could be differently affected by the presence of a tag. It would be therefore useful to the scientific community if the authors could discuss the list of controls that would be needed when applying their method to a novel target.

6) The authors assume that on average a cycling cell is halfway through the cell cycle, and that on average cells therefore have 3 copies of the genome. For mES cells which are mainly in S phase, and whose G1 and G2 are both short, this assumption may well be correct. But the authors also measure the amount of CTCF in U2OS cells, which spend most of their time in G1, and also aren't diploid. Since the authors suggest that their method can be widely applied, it would be good if they could normalise the absolute abundance of the factor of interest relative to DNA content, for example by using appropriate staining in the FACS analysis.

---

## [Author Response]

[…] Our agreed consensus is to ask you for a revised submission based on the comments below. All three of us are of the opinion that there is potentially important and useful material in the manuscript, but that significant revision is required if it is to be published in eLife.

We thank the reviewers for the review of our manuscript and their constructive suggestions for improving it. We have undertaken comprehensive revisions and an extensive number of new experiments (large changes to main figures and 3 additional supplementary figures). Briefly, our major revisions include: (A) cross-validation of our absolute quantifications using 3 distinct methods (“in-gel” fluorescence; flow cytometry; and FCS-calibrated imaging through a collaboration with Jan Ellenberg’s lab at EMBL); (B) extensive additional control experiments for Figure 1 and Figure 2; (C) a comprehensive overhaul of the text, which better highlights the limitations and includes a more extensive review of the field and previous literature as suggested.

We believe our revised manuscript fully addresses the points raised by the reviewers, whilst at the same time presenting our results in a highly conservative and cautious manner that explicitly states many of the limitations in the main text.

Clarity/over-interpretation1) Regarding the Title of the manuscript, we didn't see which 'Architectural features' have been revealed by the work. 'Quantitative features…' would be better. Furthermore, the work does not address directly '3D organisation'. A simple accurate title would do the work justice.

We have changed the Title to “Determining cellular CTCF and cohesin abundances to constrain 3D genome models”. We believe the evidence that CTCF and cohesin are involved in regulating 3D genome organization is quite strong and knowing their abundances constrains 3D genome models. We hope this is OK, but we are happy to change it and also happy to take suggestions. We found it difficult to settle on a title that simultaneously captures all aspects of the work.

2) In the Introduction, the authors don't quite correctly cite the recent work on cohesin depletion. Schwarzer et al., did not deplete cohesin itself, but like Haarhuis et al., rather depleted a subunit of the cohesin loader complex. Both these papers should be cited appropriately.

In the interest of brevity and simplicity, we did not explicitly state that Schwarzer only indirectly depleted cohesin from chromatin by conditional knock-out of NIPBL. We have now corrected this and included a more lengthy and verbose summary of the previous results from Schwarzer, Wutz and Haarhuis on NIPBL, MAU2 and WAPL.

3) One reviewer felt that 'CTCF boundary permeability' is an inappropriate term for the Abstract without explanation of what this esoteric term means. A second reviewer was of the opinion that the work did not directly address CTCF boundary permeability as stated in the Abstract. The reviewer states: 'to determine boundary permeability, one would need to do either in vitro experiments that assess cohesin passage, or otherwise perform detailed genomic analysis of TAD boundary strength; experiments beyond the scope of the current manuscript.

We have removed all mentions of “boundary permeability” from the manuscript. We now just stick to “CTCF binding site occupancy” in the Abstract and “In the context of the loop extrusion model, this suggests that the time-averaged occupancy of an average CTCF boundary site by CTCF is ~50% (Figure 1G) – that is, an extruding cohesin will be blocked ~50% of the time at an average CTCF site in the simplest version of the loop extrusion model.” in the main text. We hope that the new phrasing is both simpler and clearer.

4) Similarly, the reviewers were concerned that the absolute numbers of Rad21 taken together with the fraction that was stably bound (previous work) was distilled into a statement concerning 'the number of cohesins that are extruding loops'. We do not understand the basis of this statement, since 'loop extrusion' is not assayed or directly addressed in the manuscript. Stably DNA-bound cohesin could be doing anything, and in other cases maybe even be doing nothing (e.g if it stops translocating at appropriately paired CTCF-bound CTCF sites). Given the data, it is premature to make assumptions on the amount of extruding cohesin complexes. The authors work beautifully determines the upper limit for this number, as we now know that are no more than a certain number of complexes per cell. The authors should phrase the text accordingly and should not say that they have 'determined the density of extruding cohesins', as they do in the Abstract and in the main text.

We thank the reviewers for pointing this out and agree that we should have more clearly stated the limitations and assumptions in the main text instead of just in the Materials and methods section and we have completely re-written this part. It remains unclear if mammalian cohesin is capable of loop extrusion and we attempted to make clear that this is an assumption by writing “In the context of the loop extrusion model” etc. But we now explicitly state that the chromatin-associated cohesin fraction can only be considered an upper bound and that extruding cohesins are “putative”. In the Abstract we now write: “Extending our previous imaging studies (Hansen et al., 2017), we estimate bounds on the density of putatively DNA loop-extruding cohesin complexes and CTCF binding site occupancy …”. And in the main text we write: “For cohesin, we previously estimated the fraction of cohesin complexes that are relatively stably associated with chromatin (~20-25 min residence time in mESC G1) and thus presumably topologically engaged to be ~40% in G1 (Hansen et al., 2017). If we take this as the upper bound of putatively “loop-extruding” cohesin complexes, we can similarly calculate the upper limit on the density of extruding cohesin molecules as ~5.4 per Mb assuming cohesin exists as a monomeric ring …”. We have further clarified and specified the text in the Materials and methods as well.

5) One thing that is missing is determination of (or discussion of) the fraction of cohesin molecules that are associated with CTCF at CTCF sites (wasn't this information potentially accessible using the methods here; for example, using co-immunoprecipitation?).

This is an important but complicated question. First, our previous *eLife* manuscript (Hansen et al., 2017) showed that CTCF and cohesin bind chromatin with residence times that differ by about an order of magnitude. Therefore, the CTCF-cohesin complex that is formed cannot be a stable complex. In that paper, we showed by CoIP and super-resolution 2-color dSTORM imaging that CTCF and cohesin indeed significantly associate with each other consistent with complex formation, but technical issues prevented us (and continue to prevent us) from accurately quantifying this. Given the dynamic nature of CTCF binding on chromatin in cells and the fact that in vitro CoIP experiments only capture a fraction of in vivo CTCF-cohesin interactions, we do not feel comfortable making a quantitative estimate of which fraction of chromatin-bound CTCF proteins are directly interacting with cohesin. We now explicitly state this limitation in the main text: “although we have previously shown that CTCF and cohesin interact as a dynamic complex (Hansen et al., 2017), we are currently unable to accurately estimate what fraction of chromatin-bound CTCF proteins are directly interacting with cohesin.”

6) Overall, the manuscript needs finessing with care to avoid over-interpretation and repetition (e.g. the statement 'In the context of the loop extrusion model, is repeated at least twice). In places it is not clear whether bound CTCF or bound cohesin is being talked about; for example, in subsection “Absolute CTCF/cohesin quantification and implications for 3D genome organization”, occupancy by what? (I presume CTCF?) – this needs to be stated explicitly.

We have comprehensively re-written most sections of the manuscript and we hope that the quality of our writing has improved. We have changed the sentence referred to by the reviewer to: “, this suggests that the time-averaged occupancy of an average CTCF boundary site by CTCF is ~50% (Figure 1G) – that is, an extruding cohesin will be blocked ~50% of the time at an average CTCF site in the simplest version of the loop extrusion model.”

Cohesin oligomers?This is a contentious and much debated issue, except for the *E. coli* SMC complex MukBEF, in which the dimeric kleisin provides the basis for the inferred dimer of dimer complexes in vivo. […] Could the authors use fluorescence fluctuation measurements (e.g. Numbers and Brightness analysis or Photon Counting Histogram analysis) on Halo-Tagged cohesin? This could also provide quantitative estimates on the fraction of cohesin molecules engaged in oligomers.

The reviewers raise several issues, which we have addressed in 3 ways: (1) we have significantly toned down our interpretation of what our CoIP experiments mean for cohesin architecture; (2) We included an extensive discussion of the previous literature on SMC complex architecture (in the Discussion section, since this inevitably gets speculative); (3) we have performed additional control experiments (Figure 2—figure supplement 1), that rule out the possibility “that the IP used in the current manuscript unspecifically pulls along Rad21”.

1) Regarding toning down, we now write “This demonstrates that Rad21 either directly or indirectly self-associates in a protein-mediated and biochemically stable manner, consistent with cohesin forming dimers or higher order oligomers in vivo. However, this observation does not implicate that cohesin dimers or oligomers are a functional state of loop-extruding cohesin complexes.” And “Thus, while these results cannot exclude that some or even a majority of mammalian cohesin exists as a single-ring (Figure 2A), they do suggest that a measureable population may exist as dimers or oligomers. Whether this subpopulation represents handcuff-like dimers, oligomers (Figure 2A), cohesin clusters (Hansen et al., 2017) or an alternative state (e.g. single rings bridged by another factor such as CTCF) will be an important direction for future studies.” And “Our CoIP results show that cohesin can exist in a dimeric and/or oligomeric state in mESCs (Figure 2). These oligomers may also be arising from cohesin clusters, which we previously observed with super-resolution microscopy (Hansen et al., 2017), or even from larger complexes that contain single ring cohesins which do not directly interact.”.

2) Regarding including a more extensive discussion of the literature: we have now included it in the Discussion section. Briefly: we believe the evidence that bacterial condensin can dimerize (Badrinarayanan, 2012, Fennell-Fezzie, 2005, Matoba, 2005, Woo, 2009) and extrude loops presumably as a handcuff dimer (Wang, 2017) is quite strong. At the same time, there is also very strong evidence that budding yeast condensin can extrude loops as a single ring (Ganji, 2017). Since mammalian cohesin is quite different from both bacterial condensin and budding yeast condensin, we believe that the only thing we can say with certainty from the previous literature is that both monomeric and dimeric SMC complexes are plausible. For budding yeast, CoIPs could not detect cohesin dimers (Hearing, 2002). For mammalian cohesin, one CoIP study reported dimers (Zhang, 2008) and another CoIP study failed to detect them (Hauf, 2005). Both used over-expressed cohesin subunits, which we previously showed can cause artifacts (Hansen et al., 2017). To the best of our knowledge, our study is the first to test for cohesin self-CoIP using endogenously tagged cohesin subunits in mammalian cells. We believe that the new Discussion section takes into account most prior work and interprets the CoIP (Figure 2) experiments in a very cautious and conservative manner.

3) Regarding the CoIP specificity, first of all we would like to point out that although we only engineered mESCs to tag one Rad21 allele with FLAG and the other with the V5 peptide, we probed self-interaction using both FLAG and reciprocal V5 pull-downs. Each pull-down was repeated with antibodies raised in two different species (i.e., mouse and rabbit) and in two different mESC clonal lines (B4 in Figure 2 and A2 in Figure 2—figure supplement 1D). The results from all these experiments are consistent with a protein-mediated cohesin self-interaction. To address the reviewers' concern that our IP protocol unspecifically pulls along Rad21 we now include extensive control experiments in Figure 2—figure supplement 1. These probe specificity and cross-reactivity of all the antibodies used in this study. One antibody (the rabbit anti-FLAG previously used in Figure 2F and now replaced with a mouse anti-FLAG) seems to pull-down non-specifically wild type, untagged Rad21, and was thus excluded from further experiments and analyses. We thank the reviewers for prompting these additional and critical controls.

Validation and assumptions1) In the section 'A simple general method for counting Halo-tagged proteins in living cells', the authors do not provide sufficient data supporting the reliability of the proposed method. The text does not discuss that abundance of proteins can change throughout the cell cycle. Therefore, the method has the same limitation as other ensemble methods, unless cells are sorted into different cell cycle stages, or single-cell microscopy is used-this could be a useful discussion point?

If our method came across as “claiming to be better than ensemble methods”, this was not our intention. It is not. Cells from bacterial to mammalian are well known to exhibit considerable cell-to-cell variation in gene expression (also referred to as “noise”), which can be due to cell cycle differences but also due to the fact that gene expression is an inherently stochastic process. We now explicitly state this limitation in the main text: “these numbers represent averages (e.g. we averaged over different cell cycle phases, and protein abundance can vary significantly between phases of the cell cycle and even between genetically identical cells).” Nevertheless, we believe that knowing the average abundance of a protein per cell is still useful.

Second, the reviewers asked for better validation of the method. To do this, we approached the Ellenberg lab, experts in using FCS-calibrated imaging for absolute protein abundance estimates. Our “in-gel” fluorescence estimate of CTCF abundance in U2OS cells was 104,900+/-14,600. Their FCS-based estimate (101 single cells; 4 replicates) was 114,600+/-10,200. Thus, two completely orthogonal methods performed on different continents by different scientists yield identical results within error. When we test the two mESC C59 and C87 Halo-CTCF cell lines using the U2OS line as a standard for flow cytometry-based quantification, we similarly get convergent results (Figure 1E-F). We believe these new experiments provide strong validation of the proposed method used in Figure 3.

Nevertheless, when we apply the method to mESC C45 Rad21-Halo, our flow cytometry (FCM) and “in-gel” estimates do differ to some extent and we explicitly state this in the main text: “For mESC C45 Rad21-Halo, the FCM estimate of 131,800 +/- 12,600 proteins/cell differed more from the “in-gel” fluorescence estimate of 86,900 +/- 35,600, but was still just within error. We speculate that this discrepancy could be due to poorer Rad21-Halo protein stability during the biochemical steps of the "in-gel" fluorescence method. We again take the mean of the two methods, ~109,400 Rad21 proteins per cell, as our final, though less certain, estimate of Rad21 abundance in mESCs.”

Finally, it has also become clear to us that using a mESC line as a standard is not optimal because embryonic stem cell culture is much more specialized (e.g. if a user is not experienced with ES cell culture, it is easy to accidentally differentiate the cells). We have therefore switched to using the U2OS C32 Halo-CTCF cell line as the standard in Figure 3 since it is validated by FCS-calibrated imaging and more convenient to grow. We note that the *Sox2* and TBP estimates have now changed significantly, but we feel much more confident in the new estimates.

2) At a minimum, the authors should apply the method described in Figure 3 to compare the copy number of two proteins that they already measured with another method. For example, the authors could use the c45 (mRAD21-Halo) cell line as a standard, and apply the method of Figure 3 to the c87 (Halo-CTCF) cell line, in order to verify that the "in-gel fluorescence" and the "live-cell" method can provide comparable estimates for the number of CTCF molecules/cell. Also, it is not clear from the text whether the authors tried to quantify the Sox2 protein number by both flow-cytometry and by microscopy. If yes, the authors should mention how the estimated protein number by these two methods differ. If not, they should only describe the experiments they actually did, and maybe comment in the Discussion section about the possible extension of the proposed method using other approaches (e.g. cell cytometry vs. live microscopy) – also considering that the dynamic range of these two methods is expected to be different.

It’s a great suggestion and we have done this. We have now used the live-cell flow cytometry (FCM) method to quantify all the mESC lines (new Figure 1E-F). For CTCF they give highly similarly results to the “in-gel” fluorescence method, but for Rad21 they differ more, still just within error. We performed 4 replicates, which are shown in Figure 1—figure supplement 3.

Regarding microscopy (instead of FCM), we just meant that it was a possibility (after all, they are just different ways of reading out fluorescence). But we agree, that since we did not perform these experiments also using microscopy, we have removed all mentions of microscopy.

We believe our new FCM experiments and our new FCS-calibrated imaging experiments have fully addressed this point by providing validation through an orthogonal method.

3) Additional control experiments should be performed in order to test the reliability of the novel methods introduced and applied in this manuscript. For example, in the text that relates to Figure 1, the authors use the fluorescent signal from a bright fluorescent ligand (JF646) to quantify the protein copy number of endogenous proteins tagged with HaloTag via genome editing. The authors claim that previous work has shown that JF646 labelling is "quantitative" in living cells, but it seems that in the cited paper (Yoon et al., 2016) the fluorescent ligand is only used in a quantitative way to estimate relative changes in expression, with no real proof of its reliability for estimating absolute numbers of proteins. Importantly, the authors should cross-validate the proposed method using an established approach in the same cell line, for example by fluorescence correlation spectroscopy or mass spectrometry. At least the authors could show what are the minimum and maximum protein copy numbers measured in the cellular population and how they relate with the dynamic range in endogenous protein expression measured for example by immunofluorescence or fluorescence cytometry on the non-labelled protein population.

We have addressed this concern in two ways. First, the reviewers suggested Fluorescence Correlation Spectroscopy (FCS) as an orthogonal method of cross-validation. To do this, we approached the Ellenberg lab, who are experts in using FCS-calibrated imaging for absolute protein abundance estimates. Our “in-gel” fluorescence estimate of CTCF abundance in U2OS cells was 104,900+/-14,600. Their FCS-based estimate (101 single cells; 4 replicates) was 114,600+/-10,200. Thus, two completely orthogonal methods performed on different continents by different scientists yield identical results within error, which we believe provides validation for the “in-gel” fluorescence method of Figure 1A-B (the Ellenberg lab separately provided extensive validation for their FCS method). Second, we have performed a titration experiment to estimate the live-cell labeling efficiency of Halo-JF_646_ (show in in Figure 1—figure supplement 1A-C). At the 500 nM concentration we used, we estimate ~>90% labeling efficiency.

Finally, in our hands, immunofluorescence is not very robust and also not very generalizable (due to antibody idiosyncrasies), but we have estimated the dynamic range from Flow Cytometry by also measuring non-labeled cells (Figure 1—figure supplement 3 and Figure 3—figure supplement 1). Based on these we estimate that for very lowly expressed proteins (below ~10,000 proteins/cell), our flow cytometry method may not be sufficiently sensitive and now explicitly state this limitation in the text: “Although this method should be generally applicable, we note that it may not be robust for very lowly expressed proteins (below ~10,000 proteins per cell; Figure 3—figure supplement 1)”. Within the biologically plausible range (below many millions of proteins per cell), we do not believe our method has an upper limit on protein abundance.

4) Did the authors check if the HaloTag sequence interfere with transcription, RNA stability, protein synthesis, protein degradation? Any of these processes can influence the protein copy number. At minimum the authors should show by quantitative Western Blotting that the amount of HaloTag-CTCF in the gene-edited cell line is identical to the amount of non-tagged CTCF in the parental cell-line.

These are all excellent points, which we addressed in our previous work (Hansen et al., 2017). But we agree that we should have referred to this in this paper and have now done so by writing “We showed that these cell lines express the tagged proteins at endogenous levels by quantitative Western blotting, (Hansen et al., 2017)”. Below, we mention some of the control experiments we presented in the 2017 paper upon which the present paper builds as a Research Advance:

Untagged and homozygously tagged protein levels for CTCF and Rad21 are similar within experimental error in both mESCs and human U2OS cells: Hansen, 2017 Figure 1—figure supplement 3B.

Tagged CTCF and cohesin CoIP as efficiently as untagged proteins: Hansen et al., 2017 Figure 1G.

Endogenously tagging both CTCF and cohesin in mESCs barely affects their chromatin binding behavior as assayed using ChIP-seq (Hansen et al., 2017 Figure 1E-F; Hansen et al., 2017 Figure 1—figure supplement 4 and Figure 1—figure supplement 5).

Endogenously tagging both CTCF and cohesin in mESCs does not affect pluripotency as assayed using a teratoma assay (Hansen et al, 2017 Figure 1—figure supplement 2) nor does it affect at the RNA level the expression of key pluripotency genes (Hansen et al., 2017 Figure 1—figure supplement 3A).

5) How can the authors be sure that all the protein is extracted and detected following cell lysis? How can the authors be sure that all the protein is labelled by the JF646 fluorescent ligand? Interestingly the authors used 500nM ligand to label cellular CTCF, while a previous paper from another group using HaloTag labelling (Mazza et al., 2012 – Supplementary Figure 1) estimated that at this concentration only about 80% of a relatively low abundancy protein such as p53 is labelled by a similar concentration of TMR. The authors could titrate the concentration of fluorescent ligand added to the cells to identify a saturating concentration. The same concerns could potentially apply whenever the methods would be used on a different protein or in a different cell line. For example, the membrane permeability to the ligand might be cell-dependent and the degradation rate of different proteins could be differently affected by the presence of a tag. It would be therefore useful to the scientific community if the authors could discuss the list of controls that would be needed when applying their method to a novel target.

To estimate the labeling efficiency using Halo-JF_646_, we performed a titration experiment as suggested by the reviewers and we estimate that we are getting at least 90% labeling (Figure 1—figure supplement 1A-C). The 2012 Mazza paper is a classic landmark paper in the SPT field, but direct comparison of the labeling efficiencies are difficult. First, that paper used transient transfection, which is known to result in huge cell-to-cell variation and also in high day-to-day transfection efficiency variation. Second, close inspection of Supplementary Figure 1G in (Mazza et al., 2012) actually shows 500 nM TMR gave the highest labeling efficiency out of all tested concentrations. Third, Mazza et al., estimates that there are only 600-4,000 p53 proteins present per cell and in the revised manuscript we now state that our robust detection lower limit is around 10,000 proteins per cell. Nevertheless, we agree that we cannot exclude slight undercounting due to incomplete labeling and we now state this: “We note that JF_646_-labeling is near-quantitative in live cells (Yoon et al., 2016); moreover, a titration experiment indicates ≥90% labeling efficiency (Figure 1—figure supplement 1A-C), though we cannot exclude slight undercounting due to incomplete labeling.” As far as we know, Halo-tag dyes work well in all tissue culture cells that have ever been tested as long as the protein is significantly expressed (we even successfully labeled live E4.5 late blastocyst mouse embryo chimeras (Mir et al., 2018)). But we do acknowledge that the “in-gel” fluorescence method may suffer from protein-specific idiosyncrasies such as poor stability, difficulty with full extraction, etc. In fact, we believe that now that we have validated the absolute abundance of Halo-CTCF in U2OS cells using 2 orthogonal methods, the “live-cell” flow cytometry (FCM) method is more robust. As suggested, we now explicitly state this: “Compared to the “in-gel” fluorescence method (Figure 1A-B), we believe this live-cell FCM method is both more convenient and robust, since it avoids cell lysis and other biochemical steps that may affect protein stability.”

6) The authors assume that on average a cycling cell is half-way through the cell cycle, and that on average cells therefore have 3 copies of the genome. For mES cells which are mainly in S phase, and whose G1 and G2 are both short, this assumption may well be correct. But the authors also measure the amount of CTCF in U2OS cells, which spend most of their time in G1, and also aren't diploid. Since the authors suggest that their method can be widely applied, it would be good if they could normalise the absolute abundance of the factor of interest relative to DNA content, for example by using appropriate staining in the FACS analysis.

First, regarding the mESCs we have now performed cell cycle phase measurements and show that indeed, an average mES cell is approximately halfway through the cell cycle: 10.2% in G1; 73.9% in S-phase; 15.9% in G2. These new experiments and results are presented in Figure 1—figure supplement 1D-G.

Regarding U2OS cells, we disagree with the reviewers; as the reviewers point out the U2OS karyotype is not well-defined. For this reason, we are careful not to estimate the average CTCF binding site occupancy for U2OS cells. We only do this for mESCs. However, this does not mean that knowing the absolute abundance of CTCF proteins per U2OS cell is useless. For example, we show that the nuclear concentration of CTCF in U2OS cells is 144.3 nM. Knowing this number may help interpret DNA binding affinities (K_d_) etc. and may also be useful for other “back-of-the-envelope” calculations. The live-cell flow cytometry absolute quantification method (Figure 3) can be used in any tissue culture cell regardless of karyotype to estimate the total number of proteins per cell. How a researcher will use this number will depend on their research question and we are not in a position to dictate or predict this. Nevertheless, we now state explicitly in the main text that: “We cannot estimate CTCF binding site occupancy and probability of blocking cohesin extrusion in U2OS cells, since these cells have a poorly defined karyotype.”